*Report*

# Nuclear PD-L1 triggers tumour-associated inflammation upon DNA damage

Naoe T Nihira [ID][1], Wenwen Wu[1], Mitsue Hosoi[1], Yukiko Togashi[1], Shigeaki Sunada [ID][2], Yasuo Miyoshi[3], Yoshio Miki[4] & Tomohiko Ohta [ID][1][✉]

## Abstract

Immune checkpoint inhibitors against PD-1/PD-L1 are highly effective in immunologically hot tumours such as triple-negative breast cancer, wherein constitutive DNA damage promotes inflammation, while inducing PD-L1 expression to avoid attack by cytotoxic T cells. However, whether and how PD-L1 regulates the DNA damage response and inflammation remains unclear. Here, we show that nuclear PD-L1 activates the ATR-Chk1 pathway and induces proinflammatory chemocytokines upon genotoxic stress. PD-L1 interacts with ATR and is essential for Chk1 activation and chromatin binding. cGAS-STING and NF-κB activation in the late phase of the DNA damage response is inhibited by PD-L1 deletion or by inhibitors of ATR and Chk1. Consequently, the induction of proinflammatory chemocytokines at this stage is inhibited by deletion of PD-L1, but restored by the ATR activator Garcinone C. Inhibition of nuclear localisation by PD-L1 mutations or the HDAC2 inhibitor Santacruzamate A inhibits chemocytokine induction. Conversely, the p300 inhibitor C646, which accelerates PD-L1 nuclear localisation, promotes chemocytokine induction. These findings suggest that nuclear PD-L1 strengthens the properties of hot tumours and contributes to shaping the tumour microenvironment.

**Keywords** cGAS-STING; DNA Damage; Inflammation; NF-κB; PD-L1
**Subject Categories** Cancer; DNA Replication, Recombination & Repair; Immunology

## Introduction

Tumour cells evade immune surveillance by expressing co-inhibitory checkpoint ligands. Programmed death-ligand 1 (PD-L1) is expressed on the cell surface of tumour cells as a ligand for programmed death-1 (PD-1) and inhibits T cell activation (Chen et al, 2022; Li et al, 2022). Therapeutic antibodies against PD-1/PD-L1, also known as immune checkpoint inhibitors (ICIs), block the binding of the receptor to its ligand and eliminate tumour cells by reactivating T cells. Although the use of ICIs has shown successful clinical results in the treatment of numerous cancer types, many cases are still resistant (Walsh et al, 2023).

ICIs are considered most effective against hot tumours, wherein PD-L1/PD-1 expression is elevated with active tumour-associated inflammation (Sharma and Allison, 2015). Aberrant DNA repair machinery, another prominent feature of hot tumours, and the resulting genomic instability are predictors of susceptibility to PD-1/PD-L1 blockade therapy (Chen et al, 2022). Persistent DNA damage response signalling and tumour-associated inflammation exhibit a strong association in cancers with genomic instability, such as triple-negative breast cancer (TNBC). TNBC is the most aggressive subtype of breast cancer with poor survival outcomes. Although ICIs are effective and have improved the prognosis of TNBC, many patients still do not respond to ICI monotherapy and require some form of combination therapy, including the use of drugs that target the DNA damage response machinery to enhance the efficacy of ICIs. Thus, elucidating the precise mechanisms that regulate DNA damage-induced inflammation and PD-L1 expression is critical for optimising treatment.

Two mechanistically distinct pathways induce tumour-associated inflammation and PD-L1 expression in response to DNA damage (Sato et al, 2019). In the early phase, several hours after DNA double-strand break (DSB) induction, ATR/Chk1 kinases, which are master regulators of DNA repair, and cell cycle checkpoints are activated by single-stranded DNA products, subsequently activating the transcription factor STAT1/3. STAT1/3 induces IRF1, which transactivates PD-L1 (Sato et al, 2017). ATR/Chk1-mediated PD-L1 expression has also been observed following exposure to other genotoxic stresses (Permata et al, 2019; Sun et al, 2018). In addition to the early events, inflammation is induced in the late phase of the DNA damage response by the cyclic GMP-AMP synthase (cGAS)-stimulator of interferon genes (STING) pathway, which recognises cytoplasmic DNA resulting from micronuclei after the cells progress through the G2/M checkpoint (Bakhoum et al, 2018; Dou et al, 2017; Harding et al, 2017; Mackenzie et al, 2017). cGAS-STING is primarily known as a sensor of exogenous DNA in the cytoplasm derived from viruses or other pathogens (Ablasser et al, 2013) which induces type I interferons and proinflammatory cytokines via the activation of transcription factors IRF3 and NF-κB, respectively (Barber, 2015; Harding et al, 2017; Mackenzie et al, 2017). Importantly, PD-L1 is

[1]Department of Translational Oncology, St. Marianna University Graduate School of Medicine, Kawasaki 216-8511, Japan. [2]Juntendo Advanced Research Institute for Health Science, Juntendo University, Tokyo 113-8421, Japan. [3]Department of Surgery, Division of Breast and Endocrine Surgery, School of Medicine, Hyogo Medical University, Nishinomiya City, Hyogo, Japan. [4]Research and Development Center for Precision Medicine, University of Tsukuba, Ibaraki 305-8550, Japan. [✉]E-mail: to@marianna-u.ac.jp

also upregulated by the type I interferons IFNα and IFNβ, in addition to type II interferon IFNγ, via JAK1/2 and STAT1/3-induced IRF1, which binds to the PD-L1 promoter (Garcia-Diaz et al, 2017; Shin et al, 2017). Thus, DNA damage not only stimulates inflammation and the immune response but also upregulates PD-L1 via two distinct pathways, promoting cancer cell invasion in the inflammatory microenvironment while protecting cancer cells from attack by the immune system. However, whether and how PD-L1, especially its nuclear fraction, regulates DNA damage response and inflammation remain unclear.

The accumulation of DNA damage has been reported to induce not only PD-L1 expression but also its translocation from the plasma membrane to the nucleus after treatment with the DSB-inducing topoisomerase II inhibitor doxorubicin (DOXO) via unknown mechanisms (Ghebeh et al, 2010). However, the physiological function of nuclear PD-L1 in response to DNA damage has currently been elusive. We have previously reported that the nuclear translocation of PD-L1 is regulated by acetylation/deacetylation of the K263 residue within the cytoplasmic domain. PD-L1 is constitutively acetylated by p300 acetyltransferase, and HDAC2-mediated deacetylation of PD-L1 at K263 allows it to bind to regulators of endocytosis, cellular trafficking, and nuclear localisation, and this action is diminished by the HDAC2 inhibitor, Santacruzamate A (Gao et al, 2020). This suggests that the nuclear fraction of PD-L1 may play a pivotal role in addition to its function in the plasma membrane. Here, we report that PD-L1 interacts with ATR and promotes Chk1 activation and retention in the chromatin after genotoxic stress. Notably, PD-L1 triggered cGAS-STING and NF-κB activation, accompanied by the production of proinflammatory chemocytokines in the late phase after DSB induction. These results suggest that PD-L1 contributes to hot tumour formation by activating tumour immunity through the NF-κB pathway in response to DNA damage.

# Results

## PD-L1 is critical for Chk1 phosphorylation in response to DNA damage

We have previously searched for PD-L1-binding proteins using mass spectrometry to characterise the intracellular functions of PD-L1 and identified ~400 molecules (Gao et al, 2020). Interestingly, these molecules included several proteins involved in the DNA damage response, including ATR and ATM, which led to the investigation of whether these proteins play a significant role in PD-L1 function in response to DNA damage. In this study, we first confirmed this interaction. Full-length (FL) PD-L1 or its cytoplasmic domain-deletion mutant (ΔCT), which fails to translocate to the nucleus from the plasma membrane, was expressed in 293T cells, and the interaction with endogenous ATR and ATM was analysed via immunoprecipitation and immunoblotting. FL-PD-L1 interacted with ATR and ATM, whereas ΔCT mutant did not, suggesting that nuclear translocation allows PD-L1 to interact with these kinases (Fig. 1A). PD-L1 was also co-immunoprecipitated with ATR in anti-ATRIP immunoprecipitates from lysates of the TNBC cell line BT549 under physiological conditions (Fig. 1B). We next analyzed whether PD-L1 co-localises with ATR. As suitable PD-L1 antibodies for immunofluorescence were not available, the PD-L1 knockout (KO) TNBC cell line MDA-MB-

231, which stably expresses of HA-tagged PD-L1, was used (Fig. 1C). Immunostaining with anti-HA antibodies showed that PD-L1 was predominantly localised in the cytoplasm and plasma membrane in formalin-fixed cells without pre-extraction (Fig. 1D). However, pre-extraction treatment of cells revealed that PD-L1 also forms nuclear foci that partially co-localise with ATR (Fig. 1E), which increase after IR exposure (Fig. 1F). The specificity of the antibody was validated by PD-L1 KO cells expressing an empty vector (Fig. 1D,E).

ATR and ATM are master DNA damage-responsive kinases that regulate cell cycle checkpoints and DNA repair pathways by phosphorylating multiple substrates, including the major downstream kinases Chk1 and Chk2. Therefore, we tested whether PD-L1 plays a critical role in phosphorylation, which is essential for the activation of these kinases, in the PD-L1-positive MDA-MB-231 cells. DNA double-strand breaks (DSBs) were induced in wild-type (WT) or PD-L1 KO MDA-MB-231 cells via IR, and the phosphorylation of ATR at Thr1989, Chk1 at Ser345, ATM at Ser1981, and Chk2 at Thr68 was analysed. Notably, the phosphorylation of Chk1was diminished in PD-L1 KO cells (Fig. 2A), whereas that of ATR at T1989, which is a hallmark of ATR activation, was not affected, suggesting that ATR retains its kinase activity but cannot phosphorylate Chk1 in PD-L1 KO cells. In contrast, the phosphorylation status of ATM and its downstream kinase, Chk2, was not affected by PD-L1 depletion under these conditions (Fig. 2A). Similar results were observed with other DNA-damaging agents, namely neocarzinostatin (NCS) and DOXO, which induce DSBs (Fig. 2B,C,E), or in the PD-L1 positive BT549 cells (Fig. EV1A,B). The inhibition of Chk1 phosphorylation in PD-L1 KO cells was reversed by the addition of exogenous PD-L1, contradicting off-target effects (Fig. 2D,E). The IR exposure did not affect the cell viability during the experiment periods (Fig. EV1C).

## PD-L1 is required for chromatin retention of Chk1 and DSB repair

To further investigate the role of PD-L1 in Chk1 function in response to DSB induction, we analysed the chromatin retention of Chk1. Under normal growth conditions, a fraction of Chk1 is localised to chromatin and is phosphorylated by ATR upon genotoxic stress. This phosphorylation activates Chk1 and triggers its release from chromatin into the cytoplasm as well as the phosphorylation of downstream targets such as Cdc25A and Cdc25C (Smits et al, 2006; Zhang et al, 2005). Conversely, DNA damage has also been reported to induce the translocation of Chk1 from the cytoplasm to the nucleus and promote its chromatin association. Chromatin association is mediated by Lys63-linked ubiquitin chains promoted by BTG3, which are subsequently removed by the deubiquitinating enzyme USP3, resulting in dissociation of Chk1 from the chromatin (Cheng et al, 2013; Cheng and Shieh, 2018)). The timing of this association and dissociation from chromatin may vary depending on conditions such as cell type and the source of DNA damage; however, in any case, Chk1 is activated in the chromatin upon DNA damage. To clarify this, the chromatin association of Chk1 was first examined in MDA-MB-231 cells after IR exposure treated with or without the ATR inhibitor VE-821 (Fig. 2F). Chk1 was detected in the chromatin fraction under unstimulated conditions, and its expression increased after IR. Interestingly, in line with the suppression of Chk1 phosphorylation by VE-821 treatment, chromatin association of Chk1 was also inhibited. These data suggest that ATR activity is required not only for Chk1 activation but also for its chromatin retention in the early phase of the DSB response. Next, we examined the effect of PD-L1 depletion on the chromatin association of Chk1 after DSB induction in

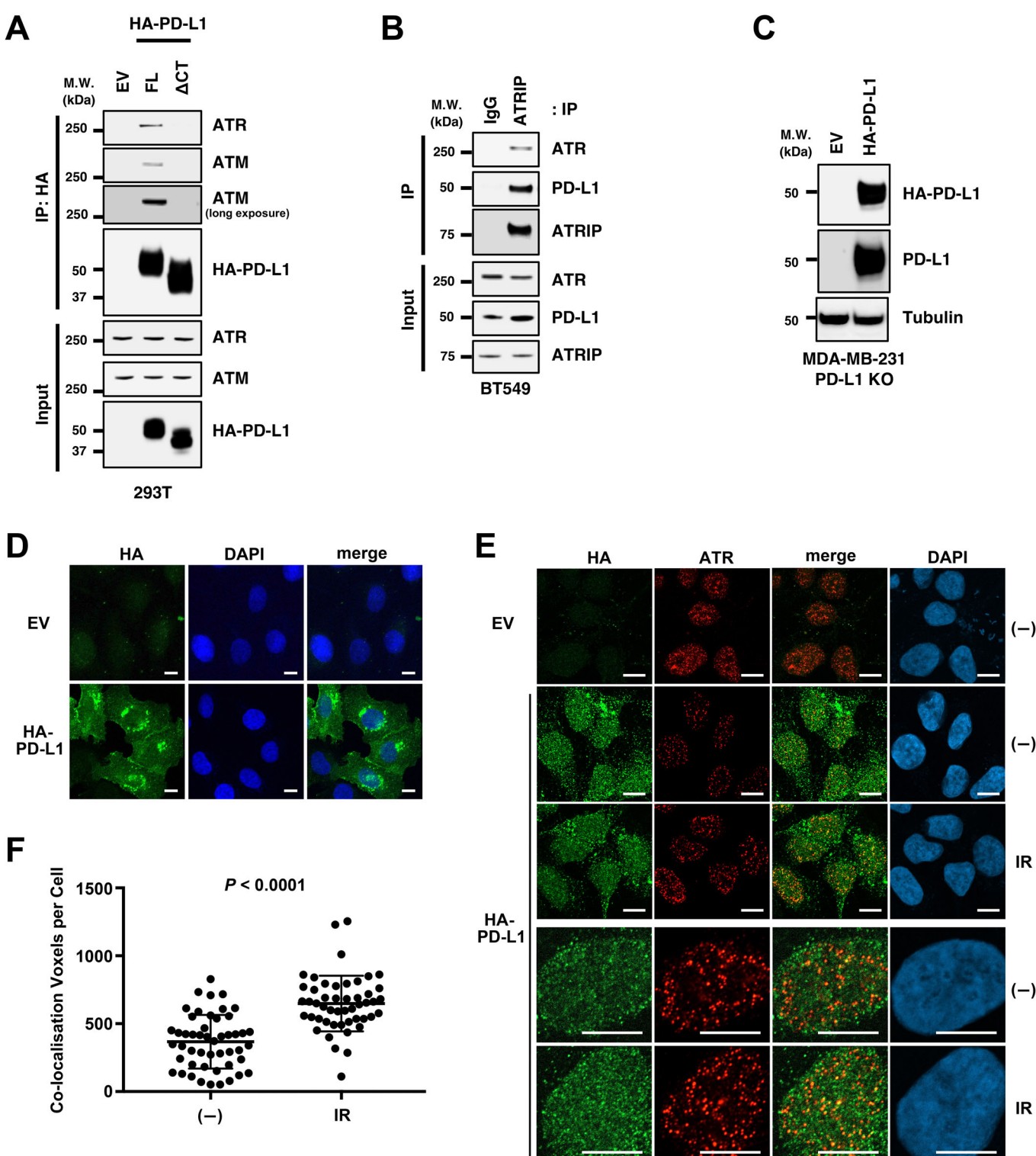

PD-L1 KO cells. Importantly, Chk1 expression in the chromatin fraction induced by IR exposure (Fig. 2G) or NCS administration (Fig. 2H) was completely abolished in PD-L1 KO cells, accompanied by inhibition of phosphorylation of another ATR substrate RPA2, and the Chk1 substrate Cdc25C. Reflecting the Chk1 inactivation, DNA replication arrest after IR was disrupted in PD-L1 KO cells (Fig. EV1D).

ATR-mediated Chk1 activation is essential for cell-cycle checkpoints and homologous recombination (HR) DSB repair (Sorensen et al, 2005). Therefore, to assess the role of PD-L1 in DSB repair, the number of nuclear γH2AX foci was measured after NCS administration and the effect of PD-L1 deletion on recovery from DSB was analysed. Although the foci steadily declined in WT cells,

◀ **Figure 1. PD-L1 interacts and co-localizes with ATR.**

(A) Lysates of 293T cells transfected with full-length (FL) or C-tail deleted (ΔCT) HA-tagged PD-L1 were immunoprecipitated with anti-HA antibody followed by western blotting. Inputs were also loaded. (B) Lysates from BT549 cells were immunoprecipitated (IP) with an anti-ATRIP antibody or preimmune IgG, followed by western blotting. Inputs were also loaded. (C, D) PD-L1 KO MDA-MB-231 cells stably expressing the empty vector (EV) or HA-tagged PD-L1 were subjected to western blotting (C) or immunofluorescence without pre-extraction (D). The nuclei were counterstained with DAPI. Scale bars: 10 μm. (E) The cells in (D) were exposed or not to 5 Gy IR; 30 min later, they were subjected to immunofluorescence with pre-extraction. Scale bars: 10 μm. (F) The co-localized foci of PD-L1 and ATR before and after IR were quantified by IMARIS software based on at least 70 cells each. Data information: All western blot data are representative of at least $n = 2$ biological replicates. In (F), data are representative of $n = 2$ biological replicates and are shown as mean ± SD with actual values. $P$ value was determined by Student's $t$ test. Source data are available online for this figure.

they remained at significantly higher levels in PD-L1 KO cells even after 24 h (Fig. EV2A), which is consistent with the role of PD-L1 in ATR/Chk1-mediated DSB repair. In support of the role of ATR/Chk1 in DSB repair, inhibition of ATR/Chk1 with VE-821 increased the sensitivity of leukemia cells to DOXO (Ghelli Luserna Di Rora et al, 2023). We confirmed that VE-821 treatment increased the sensitivity of MDA-MD-231 cells to DOXO (Fig. EV2B). Similarly, PD-L1 KO cells demonstrated significantly increased sensitivity to DOXO compared with WT cells. Importantly, VE-821 treatment did not further increase the sensitivity to DOXO in PD-L1 KO cells, indicating that PD-L1 is epistatic to ATR/Chk1 in the DSB repair function. Together, these data suggest a specific role for PD-L1 in ATR/Chk1 activation and repair of DSB, which is consistent with a recent report showing that PD-L1 promotes HR DSB repair (Kornepati et al, 2022).

## PD-L1 and ATR-Chk1 are required for DNA damage-induced NF-κB activation

DNA damage is a major stimulus that induces tumour-associated inflammation in hot tumours. NF-κB is a central component of this process. Intriguingly, NF-κB activation could be divided into two phases: acute activation, which occurs within several hours, and chronic activation, which is detected 5 to 7 days after IR and is essential for NF-κB-mediated inflammation (Kolesnichenko et al, 2021). The ATR/Chk1 axis has been shown to be involved in the NF-κB-mediated cytokine production 24 h after DOXO treatment in MDA-MB-231 cells (Carroll et al, 2016) or 7 days after IR in fibroblast cells (Kang et al, 2015). Therefore, we examined the timing of NF-κB activation based on the phosphorylation status of its subunit p65/RelA. MDA-MB-231 cells were treated with NCS, which induces only a transient DSB effect because of its extremely short half-life in culture medium, and were immunoblotted after several days of incubation (Fig. 3A,B). In line with previous reports, phosphorylated p65 was detected 1 and 5–6 days after NCS administration, with the latter timepoint showing a much higher induction. Notably, the ATR and Chk1 inhibitors VE-821 (Fig. 3A) and MK8776 (Fig. 3B), respectively, inhibited p65 phosphorylation at later timepoints. We then examined the effect of PD-L1 on Chk1 and NF-κB activation. Since phosphorylated Chk1 was not detectable at later timepoints, we examined the chromatin binding of Chk1. The chromatin binding of Chk1 observed 5–6 days after NCS administration (Fig. 3C) or IR exposure (Fig. 3D) was abrogated in PD-L1 KO cells. Supporting the role of PD-L1 in Chk1-mediated NF-κB activation, the phosphorylation of p65 at later timepoints after NCS administration (Fig. 3E) or IR exposure (Fig. 3F) was significantly inhibited by PD-L1 depletion. The IR exposure or NCS treatment did not affect the cell viability during the experiment periods (Fig. EV3A). The PD-L1-dependent phosphorylation of p65 was also observed in another TNBC cell

line, BT-549, and expression of exogenous HA-PD-L1 in PD-L1 KO cells restored induction of p65 phosphorylation, further supporting a specific role for PD-L1 in this process (Fig. EV3B).

## PD-L1 induces cGAS-STING activation

The PD-L1-dependent NF-κB activation detected at later timepoints coincided with the timing of canonical NF-κB activation via the cGAS-STING pathway, which is caused by the production and disruption of micronuclei that occur with cell cycle progression after DNA damage (Barber, 2015; Dou et al, 2017; Harding et al, 2017). PD-L1 constitutively activates cGAS-STING-IFNβ in cancer cells with chronic DNA damage (Cheon et al, 2021). Therefore, we investigated whether PD-L1 affects cGAS-STING activation after DSB induction. Phosphorylated TBK1, a marker of cGAS-STING activation, was analysed via immunoblotting. Phosphorylated TBK1 was detected 4–6 days after NCS administration (Fig. 4A) or IR exposure (Fig. 4B) in MDA-MB-231 cells, which was consistent with the timing of cGAS-STING activation. Importantly, depletion of PD-L1 inhibited TBK1 phosphorylation, which was restored when exogenous HA-PD-L1 was added, suggesting that PD-L1 is involved in the activation of cGAS-STING. The same results were observed in BT-549 cells, where other STING activation markers, phosphorylation of STING and IRF3, was also PD-L1 dependent (Fig. EV3C). Consistent with a critical role of ATR/Chk1 in the action of PD-L1, both VE-821 (Fig. 4C) and MK8776 (Fig. 4D) treatments inhibited TBK1 phosphorylation, whereas ATM inhibitor KU-55933 only exhibited mild effect (Fig. EV3D). Importantly, the addition of the ATR activator Garcinone C, which can transfer Chk1 to chromatin (Fig. EV3E), to PD-L1 KO cells restored phosphorylation of TBK1 (Fig. 4E). This suggests that the role of PD-L1 in this pathway is to activate ATR. In the cGAS-STING pathway, STING is activated by 2′3′-cyclic GMP-AMP (cGAMP), a product of the cytoplasmic DNA sensor cGAS. To elucidate the role of PD-L1 in the cGAS-STING pathway, the effect of its depletion on cGAMP production after NCS administration was analysed and found to markedly inhibit cGAMP production, which increased 5 days after NCS administration (Fig. 4F). Consistent with the recovery of TBK1 phosphorylation, NCS-induced cGAMP production, which was inhibited by PD-L1 depletion, was also restored by Garcinone C, suggesting that the main cause of cGAS-STING inactivation in PD-L1 KO cells is ATR inactivation. Baseline cGAMP level in MDA-MB-231 cells, which were relatively higher than in BT549 cells (Figs. 4F and EV3F), was also inhibited by PD-L1 KO. This could be due to higher genomic instability in this cell line. Because transfected oligo DNA dAdT was able to increase cGAMP in PD-L1 KO cells, PD-L1 likely functions upstream of cytoplasmic DNA production (Fig. EV3F). Together, these results suggest an essential role for in the cGAS-STING-dependent NF-κB activation after DSB induction.

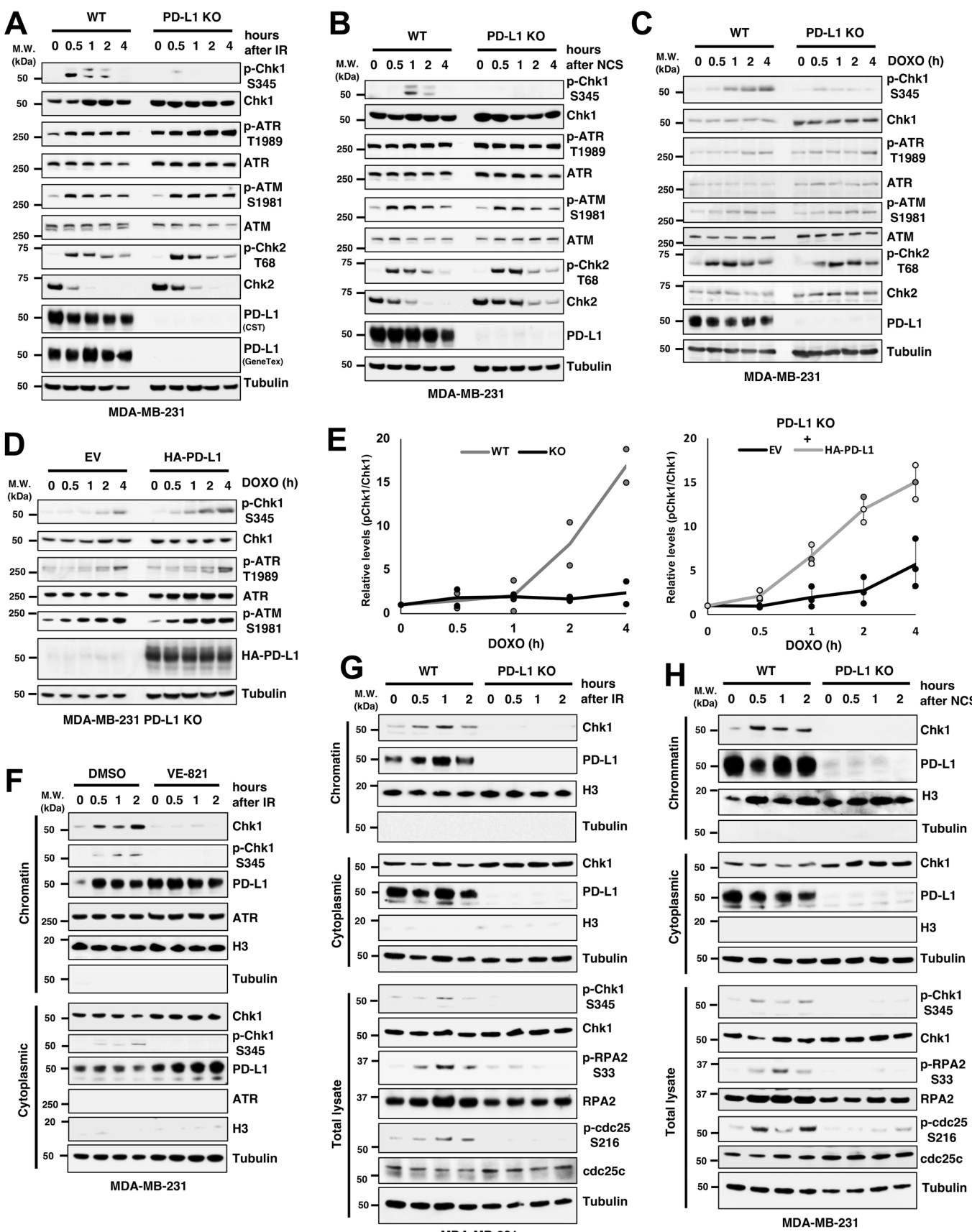

**Figure 2.  PD-L1 promotes Chk1 phosphorylation and chromatin binding after genotoxic stress.**

(A–C) WT and PD-L1 KO MDA-MB-231 cells were exposed to 10 Gy IR (A), treated with NCS (B) or DOXO (C), incubated for the indicated times and whole cell lysates were subjected to western blotting. (D) PD-L1 KO MDA-MB-231 cells transfected with empty vector (EV) or HA-PD-L1 were treated with DOXO for the indicated times and subjected to western blotting. (E) The ratio of phosphorylated Chk1 to total Chk1 shown in (C, D) was quantified from digital images using the ImageJ software. (F–H) MDA-MB-231 cells treated with vehicle DMSO or 10 μM VE-821 and exposed to 10 Gy IR (F), or WT and PD-L1 KO MDA-MB-231 cells exposed to IR (G) or treated with NCS (H) were incubated for the indicated times. Cell lysates were fractionated into chromosomal and cytoplasmic fractions and analyzed by western blotting. Fractionation efficiency was validated by the chromatin marker histone H3 (H3) and the cytoplasmic marker tubulin. Total cell lysates were also analyzed regarding the phosphorylation status of proteins. Data information: All western blot data are representative of at least $n = 2$ biological replicates. In (E), data are presented as mean with actual values from $n = 2$ (C) or $n = 3$ (D) biological replicates. Source data are available online for this figure.

## PD-L1 promotes DNA damage-induced proinflammatory chemocytokine production

The observed PD-L1-mediated NF-κB activation prompted us to investigate whether PD-L1 is also critical for the production of proinflammatory chemocytokines, which are the downstream targets of NF-κB. The mRNA expression levels of cytokines IL-6, IL-8, and GM-CSF, and chemokine Ccl-2 were analysed in NCS-treated WT and PD-L1 KO MDA-MD-231 cells. The expression of all four chemocytokines tested significantly increased 6 days after NCS administration and was markedly inhibited by PD-L1 deletion before and after NCS administration (Fig. 5A) or IR exposure (Fig. EV4A). The same results were observed in the PD-L1 KO mouse colorectal carcinoma cell line CT-26, suggesting a universal function of PD-L1 in DSB-induced inflammation, regardless of the tissue and species (Fig. 5B). Importantly, the inhibition of chemocytokine expression in PD-L1 KO cells was reversed by the addition of exogenous PD-L1, contradicting off-target effects. Reflecting the alteration in cytokine expression in the cells, the levels of IL-8 and GM-CSF secreted into the culture medium increased 5–6 days after NSC administration or IR exposure and decreased upon PD-L1 deletion (Fig. EV4B). Similar to the observation made for cGAMP, the baseline levels of chemocytokines in MDA-MB-231 cells were also suppressed by PD-L1 KO (Figs. 5 and EV4A,B). This suggests that PD-L1 mediates inflammation via endogenous DNA damage in addition to that caused by exogenous DNA damage. Consistent with the restoration of phosphorylated TBK1 and cGAMP by the ATR activator Garcinone C, NCS-induced IL-6 and GM-CSF (Fig. 5C), which were inhibited by PD-L1 depletion, were also restored by Garcinone C in a dose-dependent manner, which also suggests an essential role for ATR activation by PD-L1 in this pathway. Because transfected dAdT upregulated IL-6 in PD-L1 KO cells at similar level as that observed in WT cells, PD-L1 likely functions upstream of cytoplasmic DNA production (Fig. EV4C). Thus, the results indicate that PD-L1 promotes ATR-dependent DNA damage-induced proinflammatory chemocytokine production, in line with its role in cGAS-STING mediated NF-κB activation.

## Nuclear translocation of PD-L1 is essential for DSB-induced inflammation

Although PD-L1 is abundantly expressed on the plasma membrane, a portion internalises into the nucleus (Gao et al, 2020). In view of the finding that only full-length PD-L1, but not the cytoplasmic domain deletion mutant (ΔCT) of PD-L1, can interact with ATR (Fig. 1A), we hypothesised that the observed nuclear protein regulatory function of PD-L1 may be mediated by the nuclear fraction of PD-L1. To assess the significance of

nuclear PD-L1 for NF-κB activation, we re-evaluated the NF-κB activation status and the transcriptional levels of its target genes in PD-L1 KO cells with the addition of the ΔCT mutant. MDA-MB-231 cells with full-length or ΔCT mutants of PD-L1 were irradiated and immunoblotted with phosphorylated p65 antibody. Full-length PD-L1 but not ΔCT restored the induction of phosphorylated p65 5 days after NCS administration (Fig. 6A). Consequently, induction of IL-6 and GM-CSF was significantly reduced in cells with ΔCT PD-L1 compared with those with FL-PD-L1 (Fig. 6B). Furthermore, the effect of the mouse PD-L1 K262Q mutant, which was mutated at a site homologous to that of human K263 wherein PD-L1 cannot to translocate into the nucleus (Gao et al, 2020), was assessed in PD-L1 KO mouse CT-26 cells. The expression level of the K262Q mutant was approximately the same as in the FL cells (Fig. 6C). Similar to that observed in cells with ΔCT PD-L1, IL-6 induction was significantly reduced in K262Q mutant cells than in FL cells (Fig. 6D).

In a previous study, we revealed that K263 deacetylation by HDAC2 translocated PD-L1 into the nucleus and that HDAC2 inhibitors inhibited PD-L1 binding to DNA (Fig. EV5A) (Gao et al, 2020). Therefore, we tested the effect of the HDAC2 inhibitor Santacruzamate A on chemocytokine production after DSB induction in MDA-MB-231 cells. IL-6, IL-8 and Ccl-2 induced after NCS were all significantly reduced by Santacruzamate A, although not to the level of the effect of PD-L1 depletion (Figs. 6E and EV5B). In addition, the p300 inhibitor C646, which inhibits PD-L1 acetylation (Gao et al, 2020), accelerated the nuclear localisation of PD-L1 and enhanced the expression of IL-6 and GM-CSF in PD-L1 dependent manner (Figs. 6F and EV5C,D). These results suggest that PD-L1 translocation into the nucleus induces proinflammatory chemocytokines in response to DSBs, which can be modulated by the acetylation status of PD-L1.

## Discussion

In this study, we demonstrated that PD-L1 interacts with ATR, promotes Chk1 activation and binding to chromatin in response to DSB, and contributes to DSB repair and cellular viability. In addition, PD-L1 and ATR/Chk1 activity were required for cGAS-STING activation and the subsequent canonical activation of NF-κB and induction of its downstream proinflammatory chemocytokines in the late phase of the DSB response. The cGAS-mediated products cGAMP and cytokines, which were inhibited by PD-L1 deletion, were restored by the ATR activator Garcinone C, suggesting an ATR-dependent action of PD-L1 in this pathway. These effects of PD-L1 are prevented by genetic or pharmacological inhibition of its nuclear translocation, suggesting an important role for the nuclear fraction of PD-L1. Thus, PD-L1 not only serves as a downstream target of the immune response to DNA damage but

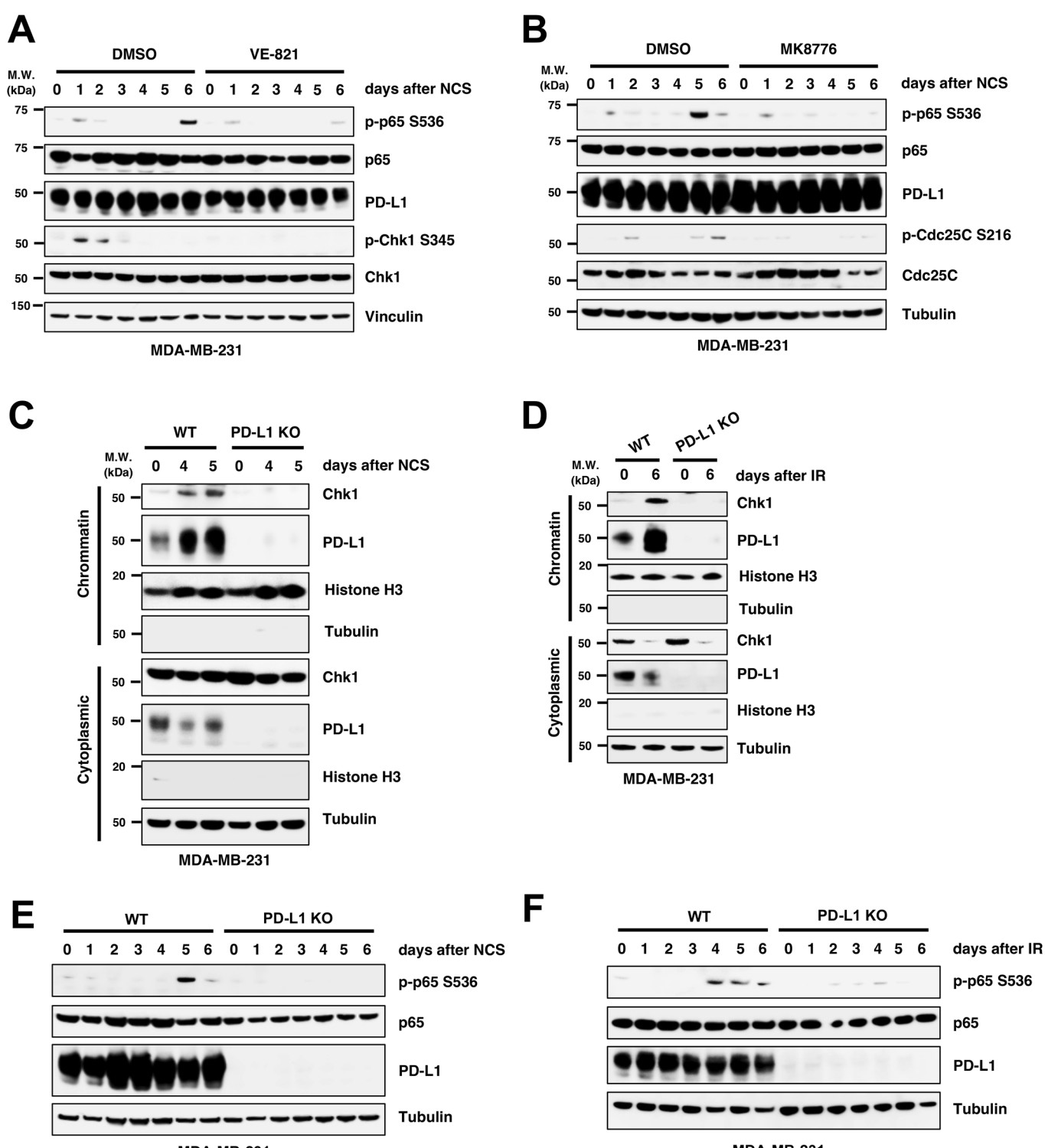

**Figure 3. PD-L1 is required for NF-κB activation in the late phase following genotoxic stress.**

(A, B) MDA-MB-231 cells were treated with NCS and cultured with either vehicle DMSO, 1 µM VE-821 (A) or MK8776 (B) for the indicated times, and whole cell lysates were analyzed by western blotting. Phosphorylated Chk1 at Ser345 (A) and phosphorylated CDC25C at S216 (B) were assessed to confirm the activity of VE-821 and MK8776, respectively. (C, D) WT and PD-L1 KO MDA-MB-231 cells untreated (day 0), treated with NCS (C) or exposed to 2 Gy IR (D) were incubated for the indicated times. Cell lysates were fractionated into chromosomal and cytoplasmic fractions and analyzed by western blotting. (E, F) WT and PD-L1 KO MDA-MB-231 cells treated with NCS (E) or exposed to 2 Gy IR (F), incubated with the indicated times were subjected to western blotting. Data information: All data are representative of $n = 2$ biological replicates. Source data are available online for this figure.

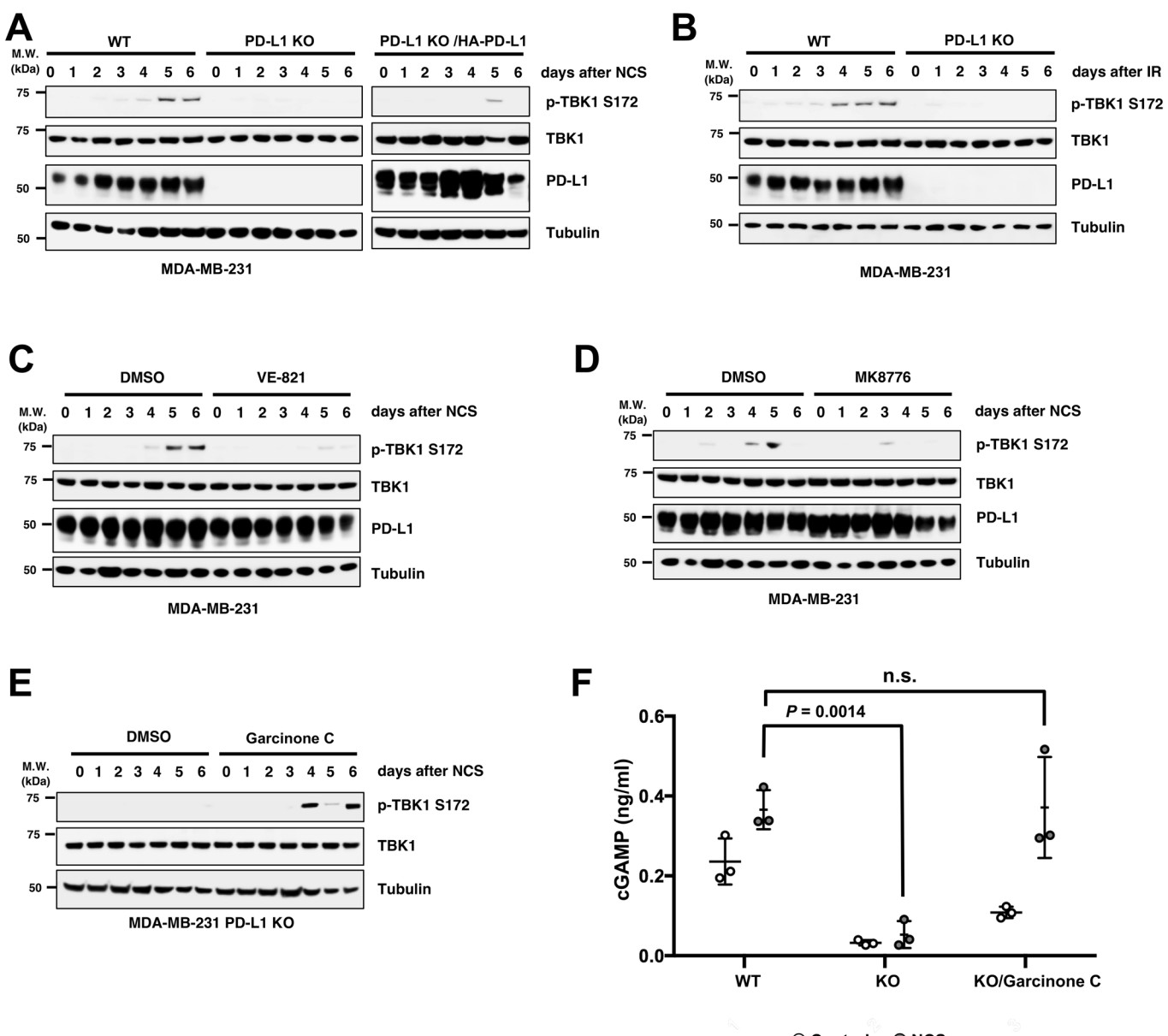

**Figure 4. PD-L1 promotes cGAS-STING activation in the late phase following genotoxic stress.**

(A, B) WT and PD-L1 KO MDA-MB-231 cells, or PD-L1 KO cells re-expressed with HA-PD-L1, as indicated, were untreated (day 0), treated with NCS (A) or exposed to 2 Gy IR (B), incubated for the indicated times and subjected to western blotting. (C, D) MDA-MB-231 cells were untreated (day 0), treated with NCS and cultured with vehicle DMSO, 1 μM VE-821 (C) or MK8776 (D) for the indicated times and subjected to western blotting. (E) PD-L1 KO MDA-MB-231 cells were untreated (day 0), treated with NCS, and cultured with vehicle DMSO or Garcinone C for the indicated times, then subjected to western blotting. (F) WT and PD-L1 KO MDA-MB-231 cells were untreated (control) or treated with NCS and cultured for 5 days with or without Garcinone C, and concentrations of cellular cGAMP were analyzed using ELISA. Data information: All western blot data are representative of at least $n = 2$ biological replicates. In (F), data representative of $n = 2$ biological replicates and are shown as mean ± SD from $n = 3$ technical replicates. $P$ values were determined by Student's $t$ test (n.s. indicates no significance). Source data are available online for this figure.

also regulates the DNA damage-induced immune response. However, the mechanism by which PD-L1 and ATR/Chk1 activate cGAS-STING remains unclear. Nonetheless, this could be partially explained by the recently discovered function of ATR in the rupture of the micronucleus envelope (Joo et al, 2023; Kovacs et al, 2023). ATR, localised to the micronucleus, phosphorylates lamin A/C in its envelope, prompting further phosphorylation by CDK1, which destabilises the micronucleus envelope. This ATR-mediated

rupture of micronuclei triggers cGAS-STING activation, cGAS-dependent autophagosome accumulation, and micronuclear DNA clearance. As our results suggest that PD-L1 acts upstream of cGAS, PD-L1 may cooperate with ATR to phosphorylate lamin A/C in the micronucleus envelope during this process.

We analysed the role of PD-L1 in the DNA damage response and inflammation induction in TNBC cells constitutively expressing PD-L1. There are three patterns of PD-L1 expression in

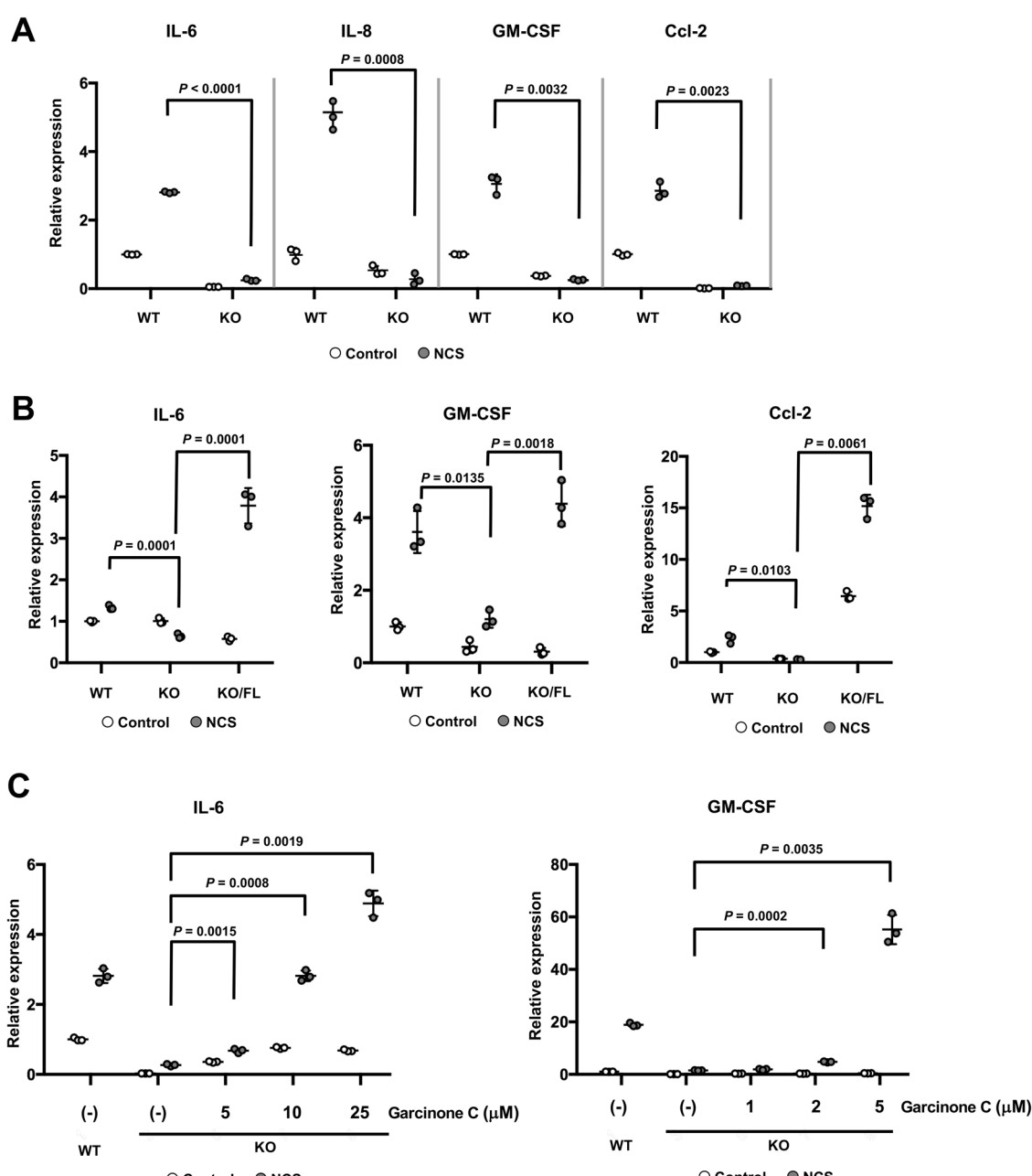

**Figure 5. PD-L1 depletion suppresses proinflammatory chemocytokines in the late phase following genotoxic stress.**

(A) WT and PD-L1 KO MDA-MB-231 cells untreated (control) or treated with NCS and incubated for 5 days were analyzed for mRNA levels of IL-6, IL-8, GM-CSF and Ccl-2 by qRT-PCR. The scores were normalized to untreated WT cells. (B) WT and PD-L1 KO mouse CT-26 cells or PD-L1 KO cells transfected with HA-PD-L1 FL (KO/FL) were untreated (control) or treated with NCS and incubated for 5 days were analyzed for mRNA levels of indicated cytokines by qRT-PCR. The scores were normalized to untreated WT cells. (C) WT and PD-L1 KO MDA-MB-231 cells were untreated (control) or treated with NCS, cultured for 5 days with or without the indicated doses of Garcinone C, and analyzed for the mRNA levels of IL-6 and GM-CSF using qRT-PCR. The scores were normalized to those of untreated WT cells. Data information: All data are representative of n = 2 biological replicates and are shown as mean ± SD from n = 3 technical replicates. P values were determined by Student's t test. Source data are available online for this figure.

cancer: (a) cancers that constitutively express PD-L1 due to genetic alterations including amplification of the *PD-L1* locus; (b) cancers that express PD-L1 induced by IFNγ, which is secreted by tumour-infiltrating T cells, in a process known as "adaptive immune resistance"; and c) cancers that do not express PD-L1, including those with a genetic event that prevents PD-L1 expression (Ribas

and Hu-Lieskovan, 2016). The effectiveness of ICIs targeting PD-1/PD-L1 depends not only on PD-L1 expression in tumour cells but also on the presence of tumour-infiltrating T cells. ICIs are not expected to be effective in cold tumours without T-cell infiltration, even in tumours that constitutively express high levels of PD-L1. Therefore, chemotherapies that induce DNA damage are used in

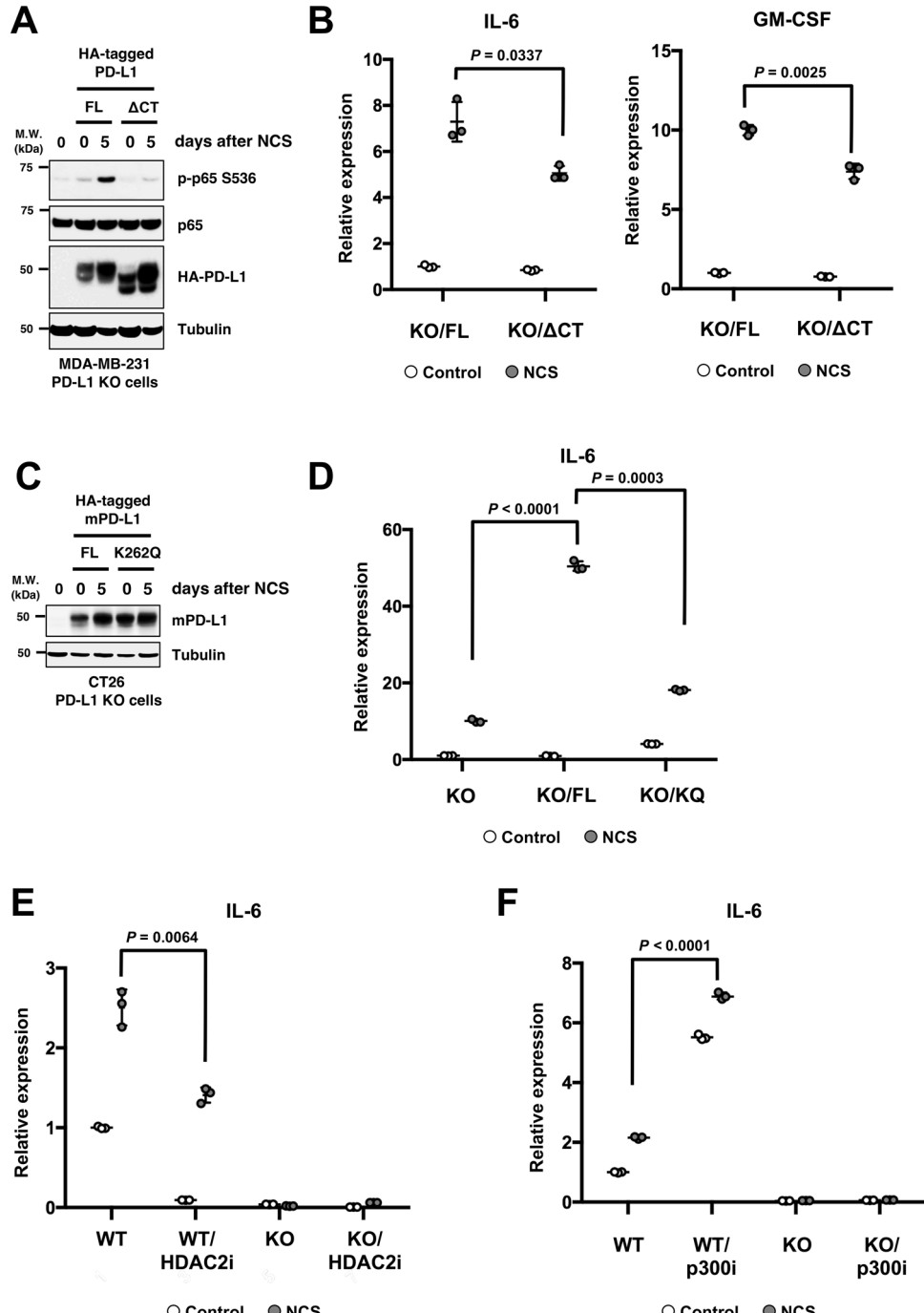

**Figure 6. PD-L1 capable of nuclear localization exclusively promotes DNA damage-induced chemocytokines.**

(A) PD-L1 KO MDA-MB-231 cells transfected with or without HA-PD-L1 FL or ΔCT were untreated (day 0) or treated with NCS and incubated for 5 days, and subjected to western blotting. (B) PD-L1 KO MDA-MB-231 cells transfected with or without HA-PD-L1 FL or ΔCT were untreated (control) or treated with NCS and incubated for 5 days, and then analyzed for mRNA levels of IL-6 and GM-CSF by qRT-PCR. The scores were normalized to untreated WT cells. (C, D) PD-L1 KO CT26 cells transfected with or without HA-tagged mouse PD-L1 FL or K262Q (KQ) mutant were untreated (day 0, control) or treated with NCS and incubated for 5 days, and subjected to western blotting (C) or qRT-PCR for IL-6 mRNA expression (D) The scores were normalized to untreated WT cells. (E, F) WT and PD-L1 KO MDA-MB-231 cells untreated (control) or treated with NCS and incubated for 5 days in the presence or absence of the HDAC2 inhibitor Santacruzamate A (E) or the p300 inhibitor C646 (F) were subjected to qRT-PCR for IL-6 mRNA expression. Data information: All western blot data are representative of at least $n = 2$ biological replicates. In (B, D–F), all data are representative of $n = 2$ biological replicates and are shown as mean ± SD from $n = 3$ technical replicates. $P$ values were determined by Student's $t$ test. Source data are available online for this figure.

combination with ICIs to convert cold tumours into hot tumours. Our results suggest that in addition to the role of PD-L1 as a target of ICI, its expression plays an important role in the inflammatory cytokine-mediated induction of T cell infiltration, that is, conversion from cold tumours to hot tumours, in such combination therapies. In contrast, in hot tumours, wherein PD-L1 expression is induced by infiltrating T cells, the expressed PD-L1 may promote the cGAS-STING pathway in response to DNA damage, which further induces its expression as well as cytokine-mediated inflammation, thus functioning as a feed-forward regulatory loop. Conversely, tumours that do not express PD-L1 may not convert to hot tumours when DNA-damaging agents are administered.

As the conversion of cold tumours into hot tumours is a promising strategy for cancer therapy, the elucidation of regulatory mechanisms of tumour microenvironment formation may uncover new approaches for cancer immunotherapies.

# Methods

## Reagents and Tools Table

| Reagent/resource | Reference or source | Identifier or catalog number |
| --- | --- | --- |
| **Experimental models** | | |
| HEK293T | ATCC | CRL-3216 |
| MDA-MB-231 | ATCC | CRM-HTB-26 |
| MDA-MB-231 PD-L1 KO | Jiao et al (2017) | Prof. Dr. Mien-Chie Hung, Taiwan |
| BT-549 | Jiao et al (2017) | Prof. Dr. Mien-Chie Hung, Taiwan |
| BT-549 PD-L1 KO | Jiao et al (2017) | Prof. Dr. Mien-Chie Hung, Taiwan |
| CT-26 PD-L1 KO | Gao et al (2020) | N/A |
| CT-26 PD-L1 KO/HA-mPD-L1 FL | Gao et al (2020) | N/A |
| CT-26 PD-L1 KO/HA-mPD-L1 K262Q | Gao et al (2020) | N/A |
| **Recombinant DNA** | | |
| HA-PD-L1 FL | Gao et al (2020) | N/A |
| HA-PD-L1 ΔCT | Gao et al (2020) | N/A |
| pLenti-HA-PD-L1 FL | Gao et al (2020) | N/A |
| **Antibodies** | | |
| Rabbit anti-PD-L1 (Human) | Cell Signaling Technology | #13684 |
| Rabbit anti-PD-L1 | GeneTex | #GTX31308 |
| Rat anti-HA | Roche | #12013819001 |
| Mouse anti-Tubulin | Merch | #T5168 |
| Rabbit anti-ATR | Cell Signaling Technology | #2790 |
| Rabbit anti-phospho ATR (Thr1989) | Cell Signaling Technology | #30632 |
| Mouse anti-Chk1 | Cell Signaling Technology | #2360 |
| Rabbit anti-phospho Chk1 (Ser345) | Cell Signaling Technology | #2348 |
| Mouse anti-Vinculin | Sigma-Aldrich | #V-4505 |
| Mouse anti-Tubulin | Sigma-Aldrich | #T-5168 |
| Rabbit anti-phospho NF-κB p65 (Ser536) | Cell Signaling Technology | #3033 |

| Reagent/resource | Reference or source | Identifier or catalog number |
| --- | --- | --- |
| Rabbit anti-NF-κB p65 | Cell Signaling Technology | #8242 |
| Mouse anti-GATA-4 | Santacruz Biotechnology | #sc-25310 |
| Rabbit anti-Histone H3 | Cell Signaling Technology | #9715 |
| Rabbit anti-ATM (Ab-3) | Merck | #PC116 |
| Rabbit anti-phospho ATM (Ser1981) | Cell Signaling Technology | #5883 |
| Rabbiit anti-phospho Chk2 (Thr68) | Cell Signaling Technology | #2197 |
| Rabbit anti-Chk2 | Cell Signaling Technology | #6334 |
| Rabbit anti-phospho Cdc25C (Ser216) | Cell Signaling Technology | #9528 |
| Rabbit anti- Cdc25C | Cell Signaling Technology | #4688 |
| Rabbit anti-cGAS | Cell Signaling Technology | #15102 |
| Rabbit anti-TBK1/NAK | Cell Signaling Technology | #38066 |
| Rabbit anti-phospho TBK1 Ser172 | Cell Signaling Technology | #5483 |
| Rabbit anti-ATRIP | Betyl laboratories | #A300-670 |
| Mouse anti-ATR | Santacruz Biotechnology | #sc-51573 |
| Rabbit anti-STING | Cell Signaling Technology | #13647 |
| Rabbit anti-phospho IRF3 (Ser396) | Cell Signaling Technology | #4947 |
| Rabbit anti-IRF3 | Cell Signaling Technology | #4302 |
| Rabbit anti-phospho RPA32/PRA2 (Ser33) | Betyl laboratories | # A300-246A |
| Rabbit anti-RPA32/PRA2 | Betyl laboratories | #300-244A |
| Mouse anti-HA (6E2) | Cell Signaling Technology | #2367 |
| Rabbit anti-ATR | Betyl laboratories | #A300-137 |
| Mouse anti-γH2AX (JBW301) | Merck Millipore | #05-636 |
| Rabbit anti-Rad51 | Bio Academia | #70-001 |
| Rabbit anti-BARD1 | Betyl laboratories | #A300-263 |
| **Oligonucleotides and other sequence-based reagents** | | |
| qRT-PCR primer | hIL-6-For | ACTCACCTCTTC AGAACGAATTG |
| qRT-PCR primer | hIL-6-Rev | CCATCTTTGGAA GGTTCAGGTTG |
| qRT-PCR primer | hIL-8-For | AAGAGAGCTCT GTCTGGACC |
| qRT-PCR primer | hIL-8-Rev | GATATTCTCTTG GCCCTTGG |
| qRT-PCR primer | hGM-CSF-For | CACTGCTGCTGA GATGAATGAAA |
| qRT-PCR primer | hGM-CSF-Rev. | GTCTGTAGGCAG GTCGGCTC |
| qRT-PCR primer | hCcl2-2-For | CAGCCAGATGCA ATCAATGCC |
| qRT-PCR primer | hCcl2-2-Rev | TGGAATCCTGAA CCCACTTCT |
| qRT-PCR primer | hIFNa-For | AGCCATCTCTGT CCTCCATGA |
| qRT-PCR primer | hIFNa-Rev | CATGATTTCTGC TCTGACAACC |
| qRT-PCR primer | hIFNb-For | GATTCCTACAAA GAAGCAGCAA |
| qRT-PCR primer | hIFNb-Rev | CAAAGTTCATCC TGTCCTTGAG |

| Reagent/resource | Reference or source | Identifier or catalog number |
|---|---|---|
| qRT-PCR primer | hActin-For | GACCTCTATGCCAACACAGT |
| qRT-PCR primer | hActin-Rev | AGTACTTGCGCTCAGGAGGA |
| qRT-PCR primer | mIL-6 For | AAGCCAGAGTCCTTCAGAGAGA |
| qRT-PCR primer | mIL-6 Rev | ACTCCTTCTGTGACTCCAGCTT |
| qRT-PCR primer | mActin For | GGCTGTATTCCCCTCCATCG |
| qRT-PCR primer | mActin Rev | CCAGTTGGTAACAATGCCATGT |
| qRT-PCR primer | mCcl-2 For | GTTGGCTCAGCCAGATGCA |
| qRT-PCR primer | mCcl-2 Rev | AGCCTACTCATTGGGATCATCTTG |
| qRT-PCR primer | mGM-CSF For | ATGCCTGTCACGTTGAATGAAG |
| qRT-PCR primer | mGM-CSF Rev | GCGGGTCTGCACACATGTTA |
| **Chemicals, enzymes and other reagents** | | |
| Poly(deoxyadenylic) acid sodium salt | Sigma-Aldrich | # P0883 |
| Santacruzamate A | Cell Signaling Technology | #69552 |
| C646 | Selleck chemicals | #S7152 |
| MK8776 | Selleck chemicals | #S2735 |
| VE-821 | Selleck chemicals | #S8007 |
| Adriamycin | Sigma-Aldrich | #D1515 |
| KU-55933 | Selleck chemicals | #S1092 |
| Neocarzinostatin | Sigma-Aldrich | #N9162 |
| Garcinone C | Med Chem Express | #HY-N695 |
| DAPI | Thermo Fisher Scientific | # P36931 |
| **Software** | | |
| GraphPad Prism 7.0 | https://www.graphpad.com | |
| ImageJ | https://imagej.nih.gov/ij/index.html | |
| Zeiss ZEN | http://www.zeiss.com/zen-lite | |
| Cellomics iDEV software | https://static.thermoscientific.com/images/D00680~.pdf | |
| IMARIS software | https://imaris.oxinst.com | |
| **Other** | | |
| Human IL-8(CXCL8) ELISA Kit | Wako | #632-42321 |
| GM-CSF ELISA Kit, Human | Proteintech | #KE00003 |
| 2'3'-cGAMP ELISA Kit | Cayman | #501700 |
| Click-iT™ EdU Cell Proliferation Kit for Imaging | Thermo Fisher Scientific | #C10337 |
| One Step TB Green® PrimeScript™ PLUS RT-PCR Kit | Takara | # RR096B |
| VARIOSKAN LUX | Thermo Fisher Scientific | |
| LAS3000 Mini | Fujifilm | |
| ImageQuant LAS-4000 | GE Healthcare | |

| Reagent/resource | Reference or source | Identifier or catalog number |
|---|---|---|
| StepOnePlus Real-time PCR system | Applied Biosystem | |
| Confocal laser scanning microscope LSM 510 or LSM710 | Carl Zeiss | |

## Cell culture, transfection, and treatments

HEK293T and MDA-MB-231 cells were cultured in Dulbecco's modified eagle medium. BT-549 and CT26 cells were cultured in Roswell Park Memorial Institute1640 medium. All media contained 10% FBS, 100 U penicillin, and 100 µg/ml streptomycin. The cells were routinely monitored for mycoplasma infection using a Mycoplasma Detection Set (Takara Bio, Shiga, Japan). MDA-MB-231, MDA-MB-231 PD-L1 KO, BT549 and BT549 PD-L1 KO cells were provided by Dr. Mien-Chie Hung (Jiao et al, 2017). CT26 PD-L1 KO cells were generated as described previously (Gao et al, 2020). Transfection was performed using FuGENE HD Transfection Reagent (Promega, Madison, WI, USA). To achieve stable expression of HA-PD-L1 in PD-L1 KO MDA-MB-231 or BT-549 cells, the cells were infected with a lentivirus containing pLenti-HA-PD-L1, which had been purified from HEK293T cells that were co-transfected with lentiviral vectors, and selected using 1 µg/mL hygromycin. The cells were treated with 1 µg/mL DOXO, 2 µM MK8776, 1 µM VE-821, 1 µM KU-55933, 5 µM Garcinone C, 25 µM Santacruzamate A, 10 µM C646 or 100 ng/mL NCS, unless otherwise stated. Information on the chemical reagents is listed in Reagents and Tools Table. For IR, the cells were exposed to X-ray irradiation at indicated doses and cultured for the indicated times before analysis.

## Plasmids

To generate HA tag-inserted PD-L1, HA sequences were inserted after the signal peptide sequence within full-length or C-tail deletion mutant (AA 263-290) PD-L1. Site-directed mutagenesis was performed via PCR and verified by sequencing.

## qRT-PCR analysis

Total RNA was isolated using a Qiagen RNeasy Mini Kit (Qiagen, Germany). qRT-PCR was performed with One Step TB Green® PrimeScript™ PLUS RT-PCR Kit (Takara Bio). The relative abundance of mRNA was calculated by normalising to the level of actin mRNA. Primer information is listed in Reagents and Tools Table.

## Immunoblot and immunoprecipitation

Cells were lysed in EBC buffer (50 mM Tris pH 7.5, 120 mM NaCl, 0.5% NP-40) containing protease (#PIA32953; Thermo Fisher Scientific, Waltham, MA, USA) and phosphatase inhibitors (#P5726; Merck, NJ, USA). The lysates were separated via sodium dodecyl sulfate-polyacrylamide gel electrophoresis and transferred to polyvinylidene difluoride membranes. The list of primary antibodies is provided in Reagents and Tools Table. Peroxidase-conjugated anti-mouse or anti-rabbit secondary antibody (#A-4416

and #A-4914; Sigma-Aldrich, St. Louis, MO, USA) were used at 1:3000 dilution. For immunoprecipitation of HA-tagged proteins, the lysates were incubated with HA agarose (#A-2095; Sigma-Aldrich). For immunoprecipitation of ATRIP, the lysates were incubated with normal rabbit IgG or an anti-ATRIP antibody (#A300-670; Betyl Laboratories, Montgomery, TX, USA) for 1 h, followed by the addition of Dynabeads and incubation for 1 h. The immune complexes were washed five times with the NETN buffer (20 mM Tris [pH 8.0], 100 mM NaCl, 1 mM EDTA, 0.5% NP-40). All the immunoblots shown in this study represent at least $n = 2$ biological replicates.

### Immunofluorescence microscopy

Indirect immunofluorescence labeling of cells and fluorescence detection without pre-extraction were performed as described previously (Zhu et al, 2021). For pre-extraction, the cells were incubated with a buffer containing 20 mM HEPES (pH 7.5), 20 mM NaCl, 5 mM $MgCl_2$, and 0.5% IGEPAL (A-630) supplemented with proteinase inhibitors and 200 μg/mL RNase A before fixation, as described previously (Wu et al, 2024). After blocking and incubation with primary and fluorescence-labeled secondary antibodies, the slides were mounted with ProLong Gold Antifade Mountant containing DAPI (Invitrogen) and examined under a confocal laser scanning microscope (LSM 510 or 710, Carl Zeiss). The colocalisation of PD-L1 and ATR was quantified using the IMARIS software (Oxford Instruments). IR-induced γH2AX foci were mechanically counted using the Cellomics iDEV software (Thermo Fisher Scientific). To assess the IR-induced replication arrest, the cells were exposed to 5 Gy IR, cultured for 4 h, and further incubated with 10 μM EdU (5-ethynyl-2′-deoxyuridine) for 15 min; subsequently, DNA synthesis was directly measured based on EdU incorporation using Click-iT EdU imaging Kits according to the manufacturer's instructions (Invitrogen).

### Clonogenic survival assay

Cells were seeded at a concentration of 1500 cells/well in 6-well plates; 24 h later, either vehicle DMSO or VE-821 was added 1 h prior to the addition of of DOXO at varying concentrations. After 6 h of incubation, the cells were washed and further cultured in fresh medium without the chemicals for 9 days. The cells were then fixed and stained with crystal violet. The colonies were scanned and counted using an ImageQuant LAS-4000 instrument (GE Healthcare).

### Purification of chromatin fraction

Chromatin and cytoplasmic fractions were purified as previously described (Cuadrado et al, 2006). Briefly, cells from a 6-cm dish were first resuspended on ice for 5 min in 60 μl of A buffer (50 mM HEPES, pH 7.4, 150 mM NaCl, 1 mM EDTA) containing 0.04% Nonidet P-40 and supplemented with protease and phosphatase inhibitors. After centrifugation, the supernatant was collected as the cytoplasmic fraction, and the pellets were washed twice with A buffer/0.04% NP-40. The pellets were then further extracted on ice for 40 min with 30 μl of A buffer containing 0.5% Nonidet P-40. After centrifugation at 160,000×$g$ for 15 min, the supernatant was collected as the chromatin-bound fraction. The lysates from both fractions were subjected to immunoblotting.

### Enzyme-linked immunosorbent assay (ELISA)

To measure the levels of secreted IL-8 and GM-SCF, cell supernatants were collected and analysed using a human IL-8 ELISA Kit (632-42321; FUJIFILM Wako Pure Chemical Corporation, Japan) or human GM-CSF ELISA Kit. To measure the level of cGAMP, cell lysates were analysed using a 2'3'-cGAMP EIA Kit (#501700; Cayman Chemical, Ann Arbor, MI, USA).

### Statistical analysis

All quantitative analyses are presented as the mean ± standard deviation of three biological or technical replicates by Student's $t$ test or two-way ANOVA using GraphPad Prism 7.0. Statistical significance was set at $P < 0.05$.

## Data availability

This study includes no data deposited in external repositories.

The source data of this paper are collected in the following database record: biostudies:S-SCDT-10_1038-S44319-024-00354-9.

## Peer review information

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

## Acknowledgements

We thank Yuka Araki for secretarial assistance. We gratefully acknowledge the support of the Center for Advanced Medical Science. This work was supported in part by the MEXT/Japan Society for Promotion of Science (JSPS) KAKENHI Grant (20H03748 and 23H02977 to TO; 23K06681 and 20J40010 to NTN). NTN is supported by the Naito Foundation (Female-42), and Research Grant of the Princess Takamatsu Cancer Research Fund (22-25426).

## Author contributions

**Naoe T Nihira**: Conceptualization; Data curation; Formal analysis; Funding acquisition; Validation; Investigation; Writing—original draft; Writing—review and editing. **Wenwen Wu**: Data curation; Formal analysis; Validation; Investigation; Methodology. **Mitsue Hosoi**: Data curation; Investigation. **Yukiko Togashi**: Data curation; Investigation. **Shigeaki Sunada**: Methodology. **Yasuo Miyoshi**: Conceptualization; Supervision. **Yoshio Miki**: Conceptualization; Supervision. **Tomohiko Ohta**: Conceptualization; Resources; Supervision; Funding acquisition; Investigation; Methodology; Writing—original draft; Project administration; Writing—review and editing.

Source data underlying figure panels in this paper may have individual authorship assigned. Where available, figure panel/source data authorship is listed in the following database record: biostudies:S-SCDT-10_1038-S44319-024-00354-9.

## Disclosure and competing interests statement

The authors declare no competing interests.

# Expanded View Figures

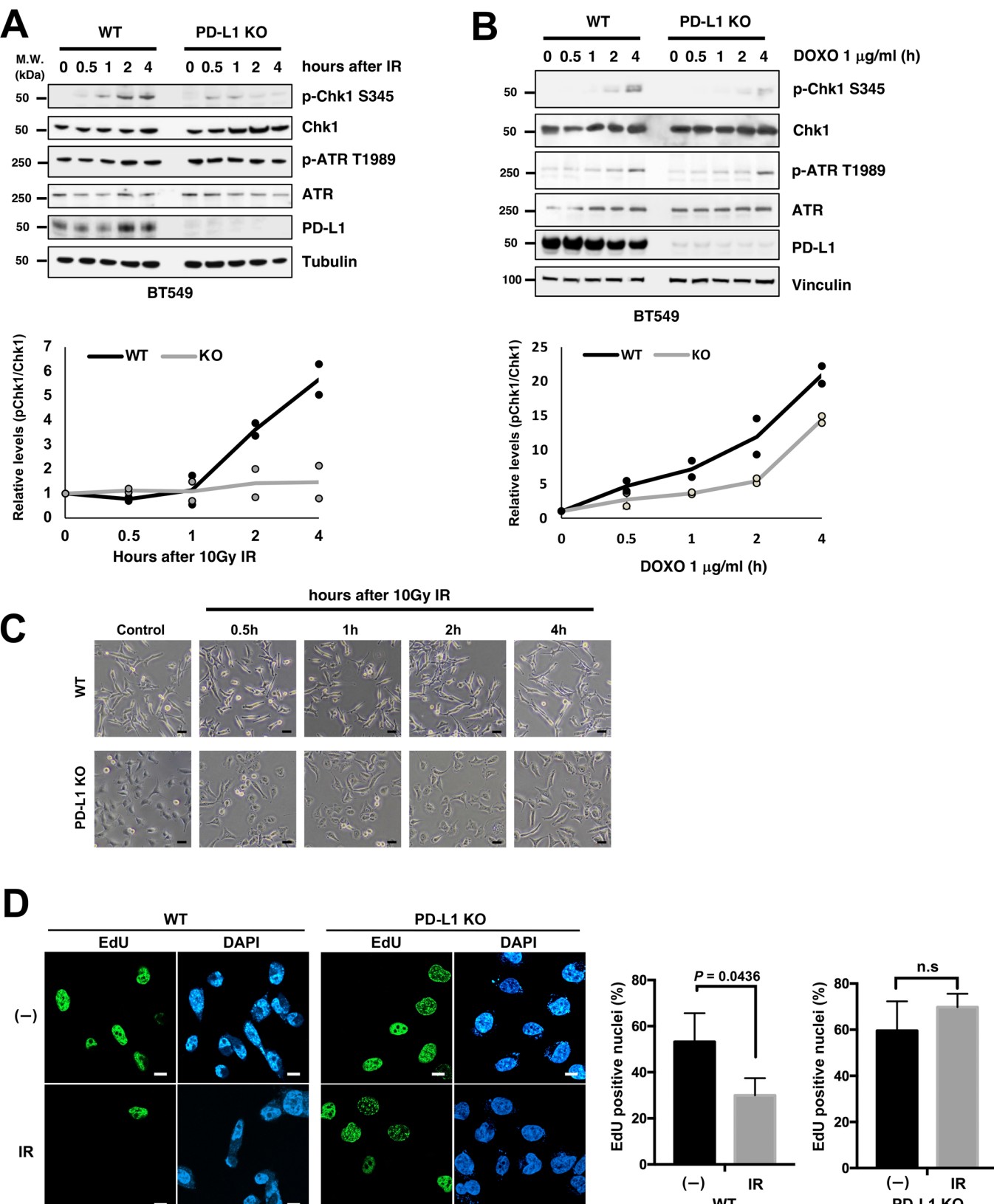

**Figure EV1.  PD-L1 is involved in ATR-Chk1 signalling pathway.**

(A, B) WT or PD-L1 KO BT549 cells exposed to 10 Gy IR and incubated for the indicated times (A) or treated with DOXO for the indicated times (B) were subjected to western blotting. The lower panels depict the ratio of phosphorylated Chk1 to total Chk1, as quantified from digital images derived from $n = 2$ biological replicates using the ImageJ software. (C) Phase contrast findings of WT or PD-L1 KO MDA-MD-231 cells exposed to 10 Gy IR and incubated for the indicated times. Scale bars: 20 μm. (D) WT or PD-L1 KO MDA-MB-231 cells were exposed or not to 5 Gy IR and incubated with EdU for 4 h. Cells that incorporated EdU were examined under a confocal laser scanning microscope. Scale bars: 10 μm. The right panels depict the quantification of the EdU-positive nuclei. The number of EdU-positive cell is shown below the images. Data information: In (A, B), data are presented as mean with actual values from $n = 2$ biological replicates. In (D), data are presented as mean ± SD from $n = 3$ biological replicates, each based on more than 50 cells. P values were determined by Student's t test (n.s. indicates no significance). Source data are available online for this figure.

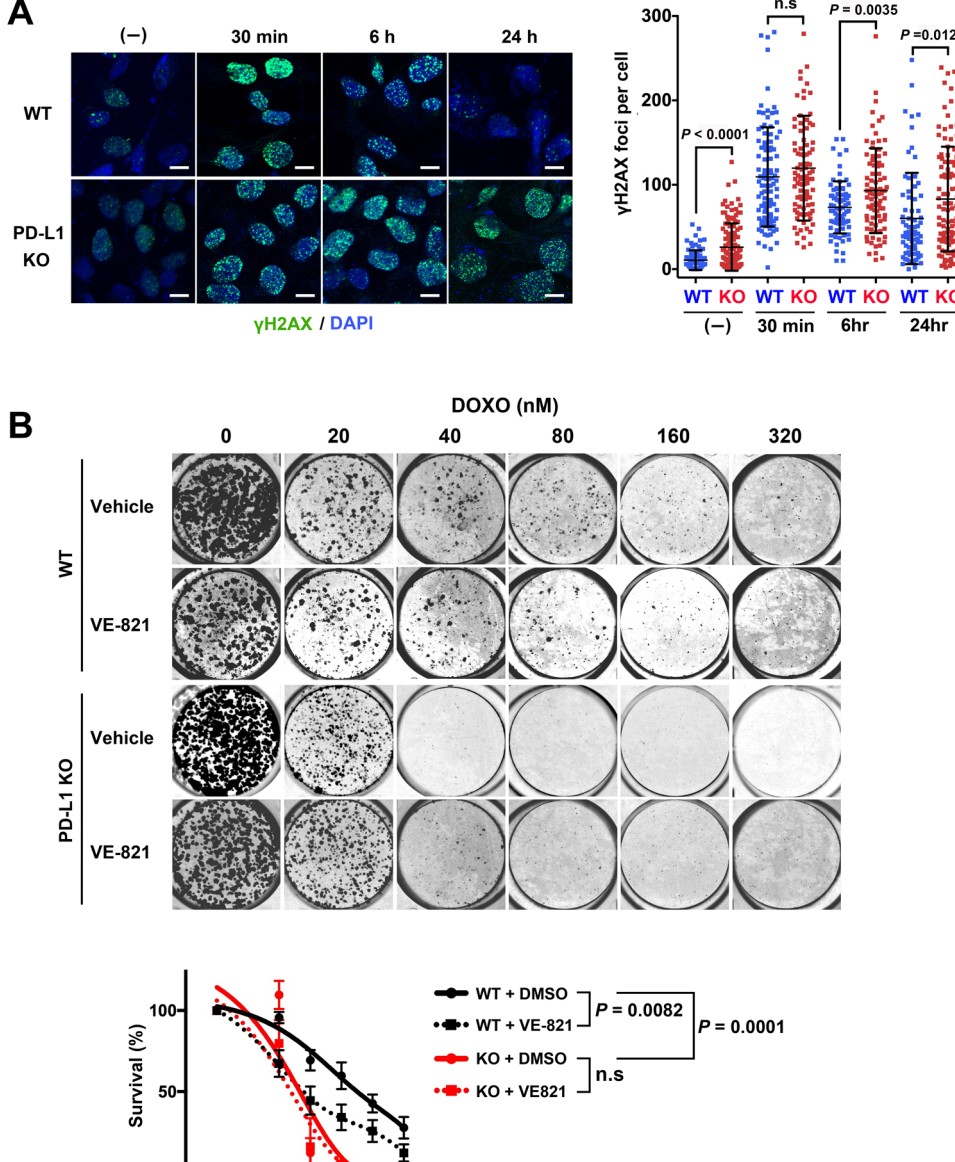

**Figure EV2.  PD-L1 depletion increases DNA damage and sensitizes cells to DSB agents.**

(A) WT or PD-L1 KO MDA-MB-231 cells were treated with or without NCS for 15 min, washed, cultured for the indicated time, and subjected to immunofluorescence using the anti-γH2AX antibody. Nuclei were counterstained with DAPI. The number of nuclear γH2AX foci per cell is shown in the right panel. Scale bars: 10 μm. (B) WT and PD-L1 KO MDA-MB-231 cells were treated with the indicated doses of DOXO in the presence of vehicle DMSO or VE-821 for 24 h, then analyzed for clonogenic survival. The relative survival data are shown below the images. Data information: In (A), data are representative of $n = 2$ biological replicates and are shown as mean ± SD with actual values. P values were determined by Student's $t$ test (n.s. indicates no significance). In (B), data are presented as mean ± SD from $n = 3$ biological replicates. Statistical significance was calculated using a two-way ANOVA (n.s. indicates no significance). Source data are available online for this figure.

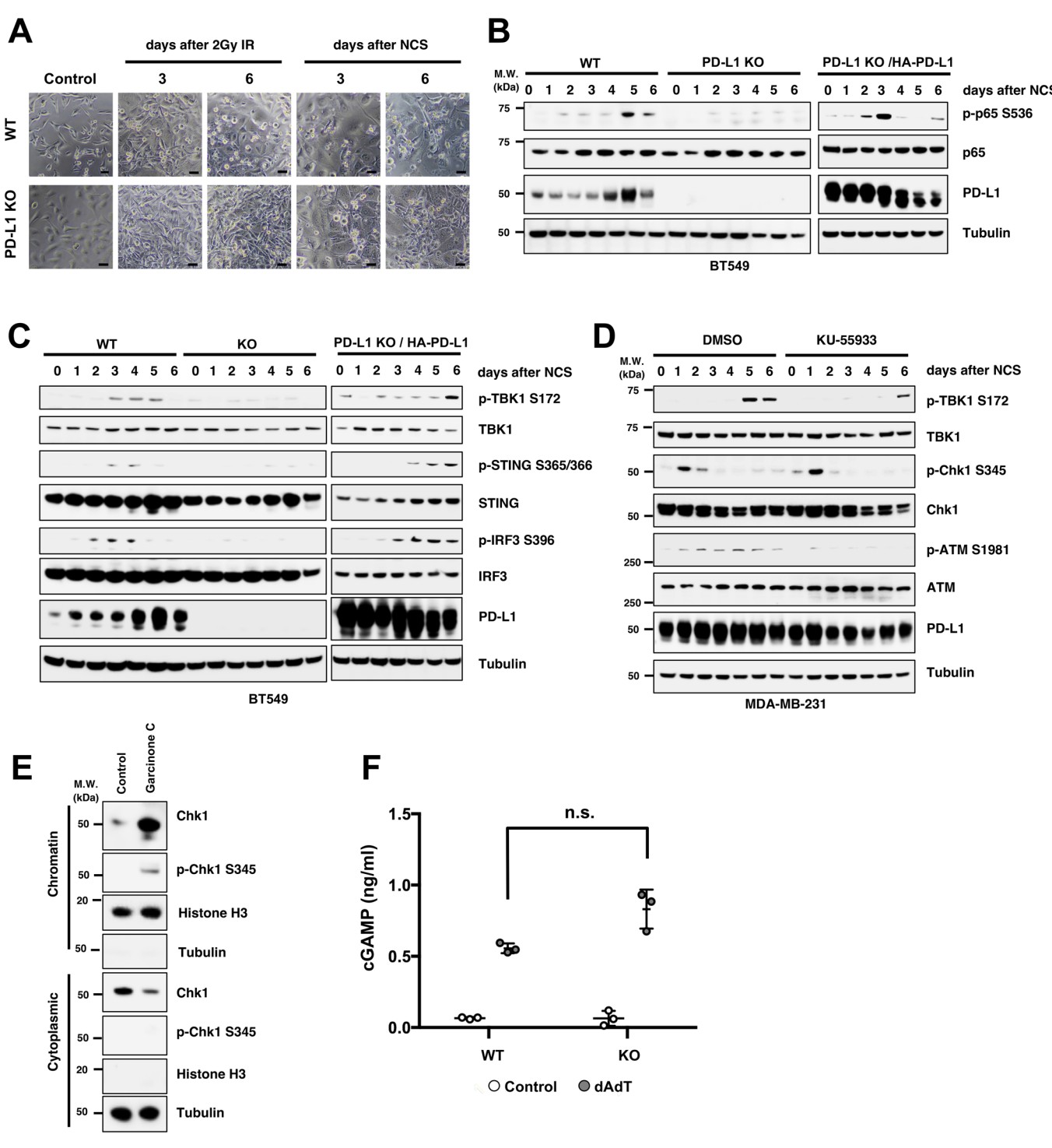

**Figure EV3. PD-L1 promotes NF-κB and cGAS-STING activation following genotoxic stress.**

(A) Phase contrast findings of WT or PD-L1 KO MDA-MD-231 cells exposed to 2 Gy IR and incubated for the indicated times. Scale bars: 20 μm. (B, C) WT and PD-L1 KO BT549 cells, or PD-L1 KO cells re-expressing HA-PD-L1 were untreated (day 0) or treated with NCS, incubated for the indicated time, and subjected to western blotting. (D) MDA-MB-231 cells untreated (day 0) or treated with NCS were cultured with vehicle DMSO or KU-55933 for the indicated time and subjected to western blotting. (E) PD-L1 KO MDA-MB-231 cells were treated with DMSO (Control) or Garcinone C for 24 h. Cells were fractionated into chromatin and cytoplasmic fractions and analyzed by western blotting. (F) WT and PD-L1 KO BT549 cells were transfected with or without oligo DNA dAdT and incubated for 24 h, and the concentrations of cellular cGAMP were analyzed using ELISA. Data information: All western blot data are representative of at least $n = 2$ biological replicates. In (F), data are representative of $n = 2$ biological replicates and are shown as mean ± SD from $n = 3$ technical replicates. $P$ value was determined by Student's $t$ test (n.s. indicates no significance). Source data are available online for this figure.

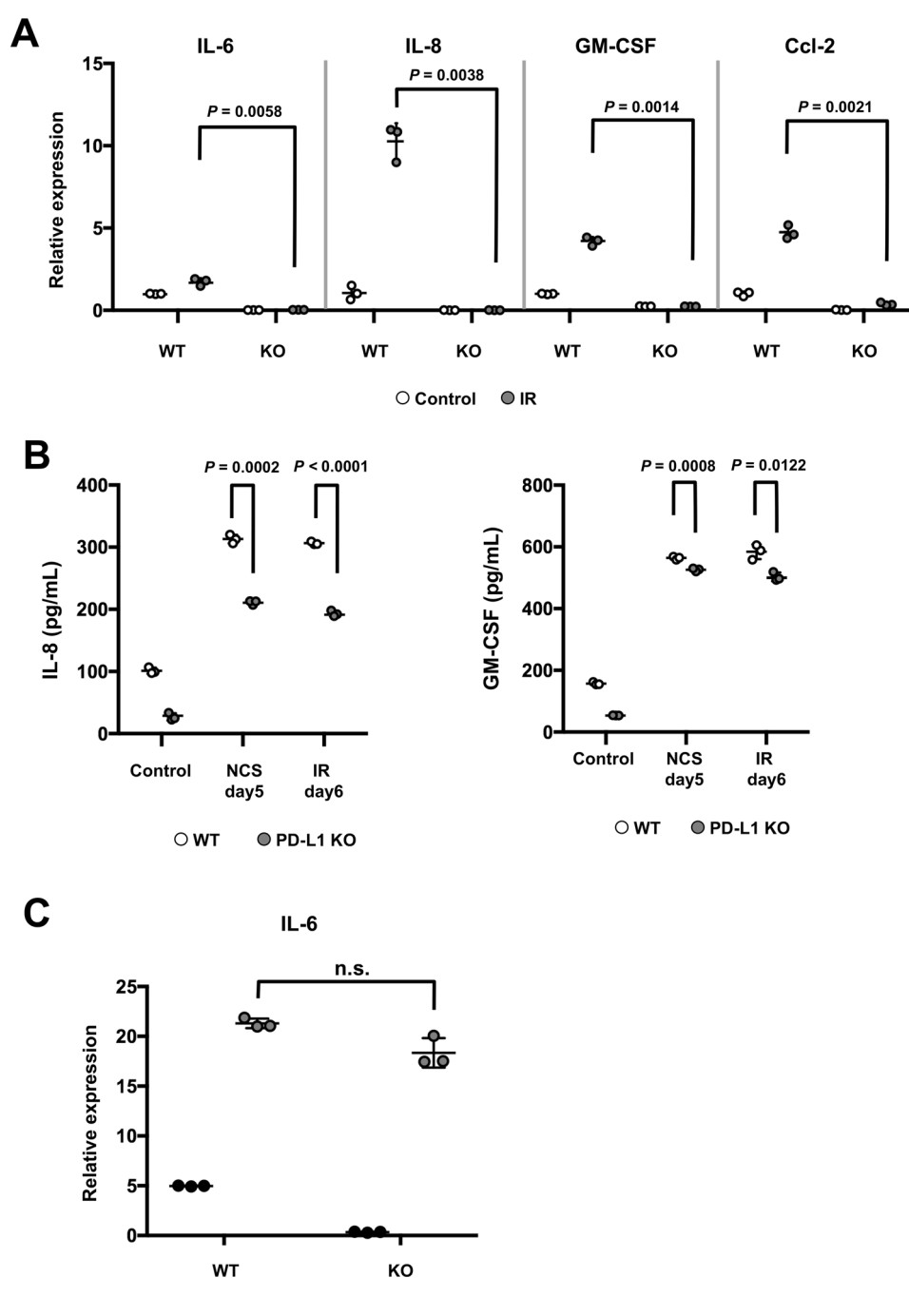

**Figure EV4. PD-L1 depletion suppresses proinflammatory chemocytokines in the late phase following genotoxic stress.**

(A) WT or PD-L1 KO MDA-MB-231 cells exposed to 2 Gy IR and cultured for 6 days were analyzed for mRNA expression levels of IL-6, IL-8, GM-CSF and Ccl-2 by qRT-PCR. The scores were normalized to untreated WT cells. (B) WT and PD-L1 KO MDA-MB-231 cells were untreated (control), or treated with NCS or exposed to IR and incubated for 5 or 6 days, respectively, and the concentrations of secreted IL-8 and GM-CSF in culture medium were analysed using ELISA. (C) WT and PD-L1 KO BT549 cells were non-transfected or transfected with oligo DNA dAdT, incubated for 24 h, and analyzed for the mRNA expression levels of IL-6 using qRT-PCR. The scores were normalized to those of untreated WT cells. Data information: All data are representative of $n = 2$ biological replicates and are shown as mean ± SD from $n = 3$ technical replicates. $P$ values were determined by Student's $t$ test (n.s. indicates no significance). Source data are available online for this figure.

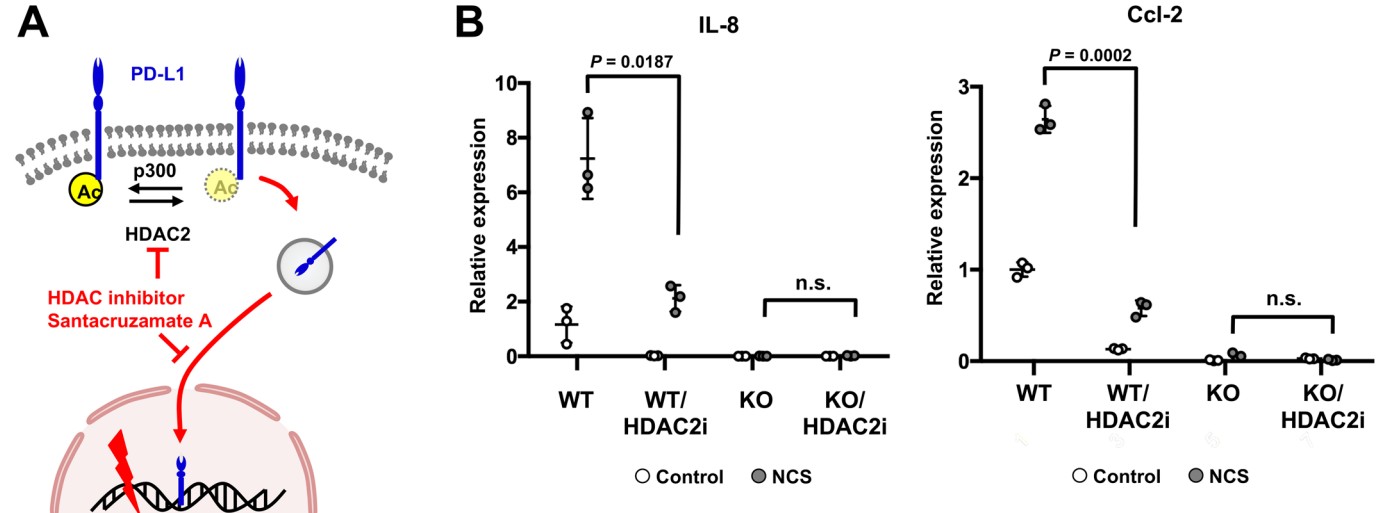

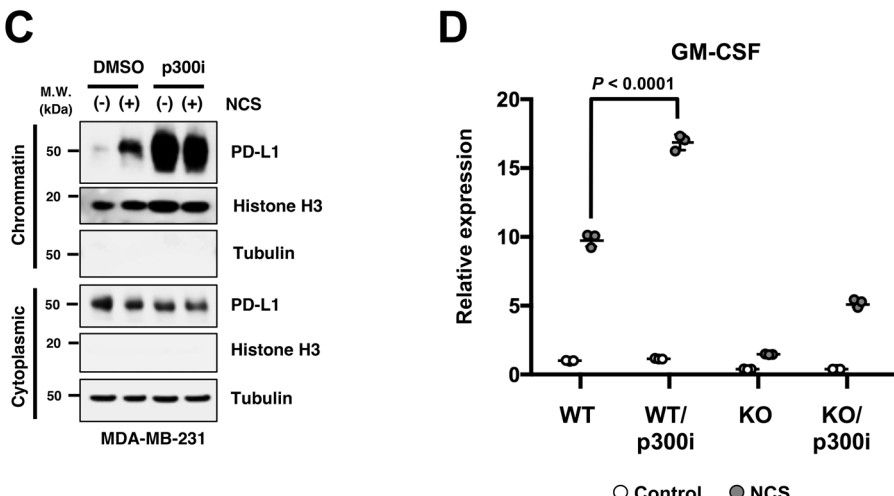

**Figure EV5.  Nuclear translocation of PD-L1 is mediated by HDAC2.**

(A) The diagram depicts a role of HDAC2 in nuclear translocation of PD-L1 (Gao et al, 2020). PD-L1 expressing on the plasma membrane is constitutively acetylated by p300 on its C-terminus. HDAC2-mediated deacetylation of PD-L1 triggers its translocation into the nucleus from the membrane. HDAC2-specific inhibitor Santacruzamate A blocks this nuclear translocation. (B) WT and PD-L1 KO MDA-MB-231 cells untreated (control) or treated with NCS and incubated for 5 days in the presence or absence of the HDAC2 inhibitor Santacruzamate A were subjected to qRT-PCR for the indicated chemocytokines. (C) MDA-MB-231 cells were untreated (−) or treated with NCS (+) and incubated for 5 days in the presence or absence of the p300 inhibitor C646. Cell lysates were fractionated into chromosomal and cytoplasmic fractions and analyzed by western blotting. Data are representative of $n = 2$ biological replicates. (D) WT and PD-L1 KO MDA-MB-231 cells untreated (control) or treated with NCS and incubated for 5 days in the presence or absence of the p300 inhibitor C646 were subjected to qRT-PCR for the indicated chemocytokines. Data information: In (B, D), data are representative of $n = 2$ biological replicates and are shown as mean ± SD from $n = 3$ technical replicates. $P$ values were determined by Student's $t$ test (n.s. indicates no significance). Source data are available online for this figure.

