## [Peer Review File · EMBO Reports]

Nuclear PD-L1 triggers tumour-associated inflammation upon DNA damage

Naoe Nihira, Wenwen Wu, Mitsue Hosoi, Yukiko Togashi, Shigeaki Sunada, Yasuo Miyoshi, Yoshio Miki, and Tomohiko Ohta

Corresponding author(s): Tomohiko Ohta (to@marianna-u.ac.jp)

Review Timeline:

Submission Date:	25th Jun 24
Editorial Decision:	5th Aug 24
Revision Received:	4th Nov 24
Editorial Decision:	28th Nov 24
Revision Received:	2nd Dec 24
Accepted:	8th Dec 24

Editor: Achim Breiling

Transaction Report:

Dear Prof. Ohta,

Thank you for the submission of your manuscript to EMBO reports. I have now received the reports from the three referees that were asked to evaluate your study, which can be found at the end of this email.

As you will see, the referees find the study interesting, but they also have several comments, concerns, and suggestions, indicating that a major revision of the manuscript is necessary to allow publication of the study in EMBO reports. As the reports are below, and all the concerns need to be addressed, I will not detail them further here.

Acceptance of your manuscript will depend on a positive outcome of a second round of review. It is EMBO reports policy to allow a single round of revision only and acceptance of the manuscript will therefore depend on the completeness of your responses included in the next, final version of the manuscript.

1) a .docx formatted version of the final manuscript text (including legends for main figures, EV figures and tables), but without the figures included. Figure legends should be compiled at the end of the manuscript text.

2) individual production quality figure files as .eps, .tif, .jpg (one file per figure), of main figures and EV figures. Please upload these as separate, individual files upon re-submission.

4) a complete author checklist, which you can download from our author guidelines

(<https://www.embopress.org/page/journal/14693178/authorguide>). Please insert page numbers in the checklist to indicate where the requested information can be found in the manuscript. The completed author checklist will also be part of the RPF.

5) that primary datasets produced in this study (e.g. RNA-seq, ChIP-seq, structural and array data) are deposited in an appropriate public database. If no primary datasets have been deposited, please also state this in a dedicated section (e.g. 'No primary datasets have been generated and deposited'), see below.

The accession numbers and database should be listed in a formal "Data Availability" section (placed after Materials & Methods) that follows the model below. This is now mandatory (like the COI statement). Please note that the Data Availability Section is restricted to new primary data that are part of this study. This section is mandatory. As indicated above, if no primary datasets have been deposited, please state this in this section

Data availability

8) Regarding data quantification and statistics, please make sure that the number "n" for how many independent experiments were performed, their nature (biological versus technical replicates), the bars and error bars (e.g. SEM, SD) and the test used to calculate p-values is indicated in the respective figure legends (also for EV figures and all those in an Appendix). Please also check that all the p-values are explained in the legend, and that these fit to those shown in the figure. Please provide statistical testing where applicable. Please avoid the phrase 'independent experiment', but clearly state if these were biological or technical replicates. Please also indicate (e.g. with n.s.) if testing was performed, but the differences are not significant. In case n=2, please show the data as separate datapoints without error bars and statistics. See also: <http://www.embopress.org/page/journal/14693178/authorguide#statisticalanalysis>

9) Please add scale bars of similar style and thickness to microscopic images, using clearly visible black or white bars (depending on the background). Please place these in the lower right corner of the images themselves. Please do not write on or near the bars in the image but define the size in the respective figure legend.

10) Please also note our reference format:

12) We now use CRediT to specify the contributions of each author in the journal submission system. CRediT replaces the author contribution section. Please use the free text box to provide more detailed descriptions and do NOT provide your final manuscript text file with an author contributions section. See also our guide to authors: <https://www.embopress.org/page/journal/14693178/authorguide#authorshipguidelines>

13) All Materials and Methods need to be described in the main text using our 'Structured Methods' format, which is required for all research articles. According to this format, the Materials and Methods section should include a Reagents and Tools Table (listing key reagents, experimental models, software, and relevant equipment and including their sources and relevant

identifiers), uploaded as separate file, followed by a Methods and Protocols section in which we encourage the authors to describe their methods using a step-by-step protocol format with bullet points, to facilitate the adoption of the methodologies across labs. More information on how to adhere to this format as well as downloadable templates (.doc) for the Reagents and Tools Table can be found in our author guidelines (section 'Structured Methods'):

14) Please order the manuscript sections like this, using these names:

Title page - Abstract - Keywords - Introduction - Results - Discussion - Methods - Data availability section - Acknowledgements - Disclosure and Competing Interests Statement - References - Figure legends - Expanded View Figure legends

I look forward to seeing a revised form of your manuscript when it is ready.

Yours sincerely,

Referee #1:

The submitted manuscript addresses the question to what extent PD-1/PD-L1 has a direct effect on the regulation of the DNA damage response after genotoxic stress. The authors observed that nuclear PD-L1 activates the ATR-Chk1 signaling pathway upon genotoxic stress and regulates the expression of proinflammatory chemokines. They observed that PD-L1 interacts with ATR and is essential for Chk1 activation and chromatin binding. The cGAS-STING and NF- κ B activation in the late phase of DNA damage response was inhibited by PD-L1 deletion or by inhibitors of ATR and Chk1. Consequently, induction of proinflammatory chemocytokines at this phase was inhibited by deletion of PD-L1 or its mutation, which inhibits nuclear localization, or by the HDAC2 inhibitor santacruzamate A, which blocks PD-L1 deacetylation and nuclear localization. Conversely, chemocytokine induction was enhanced by the p300 inhibitor C646, which accelerated PD-L1 nuclear localization.

These results suggest that nuclear PD-L1 can directly enhance DNA damage repair and thus could also influence the tumor's intracellular response to immune checkpoint inhibitors.

The manuscript deals with a biologically important mechanism whose elucidation could be of great importance for tumor therapy. I therefore consider the manuscript to be very relevant and interesting. However, I do not consider it to be sufficiently mature in its current form. I would therefore like to make a few suggestions for further improvement of the manuscript before I could agree to its publication in EMBO reports.

As the authors themselves note in their discussion, the most direct evidence of the effect of PD-L1 on the phosphorylation of CHK1 would be on the initiation of the cell cycle checkpoint, the activation of dormant replication origins, the slowing down of replication, the accumulation of RPA foci, to name just a few very important mechanisms. Since the further experiments are based on this hypothesis, I consider it necessary that at least two of these processes are demonstrated by the authors.

Furthermore, the functions of ATR and ATM cannot be strictly separated from each other. It would therefore make sense to carry out at least one of the experiments with an ATM inhibitor for comparison.

It is not clear why etoposide was used for the induction of DNA damage. The data shown so far are based on the DNA-damaging effect of irradiation, Neocarzinostatin and Doxorubicin. Would it therefore make more sense to investigate their effect on the formation of DNA damage? Or, conversely, also show the effect of etoposide treatment on CHK1 activation, chromatin binding and proliferation activity?

The authors extensively describe the importance of CHK1 for the formation of RAD51 and the recruitment of BRCA1, but show no data on whether this is impaired in their system. This would be very interesting, especially in comparison to the HA-PD-L1 deletion mutant.

I also do not consider the exclusive use of the CellTiterGlo assay to describe the effects on cell survival to be sufficient. The data obtained in this way only indicate that the cells were inhibited in proliferation for 48 hours, but do not provide any information about cell death. At least a colony formation test or an adequate test to determine survival after a longer period of time would be necessary here.

Minor points:

It would be useful to include data from different experiments in the main manuscript, such as DNA damage or survival. The manuscript consists primarily of Western blots and mRNA expression data. This is very difficult to read and understand. For protein expression, graphs with quantifications would also be useful for easier comparability.

The authors describe that for the CellTiterGlo assay 800 cells were seeded in 96-well plates; it would be more interesting to know how many cells were seeded per well.

Figure 1B: The irradiation dose should be indicated either in the figure or its legend. And wouldn't at least one more dose point make sense, since at 10 Gy a relatively large amount of cells should already be dead?

Figure 2A: p-p65 S536? Is this a control? And wouldn't it make more sense to show this at the bottom of the figure so that the important data can be seen immediately?

Referee #2:

The authors of this manuscript reported a new function of nuclear PD-L1 in regulating DNA damage response pathway the ATR-Chk1 signaling and inducing proinflammatory chemo cytokines genotoxic stress. PD-L1 is found to interact with ATR through a proteomic analysis and consequently required for Chk1 activation. It is interesting that PD-L1 depletion inhibits DNA damage - triggered cGAS-STING and NF-kappaB activation in a similar manner as inhibition of ATR and Chk1 activation. Furthermore, experiment data showed that altered PD-L1 localization in nuclear by the HDAC2 inhibitor (reduced localization) or the p300 inhibitor (enhanced localization) changed the induction of proinflammatory chemocytokines in the late phase of DNA damage response. Collectively, these findings uncovered a new role of nuclear PD-L1 in regulating DNA damage-triggered immune responses, which potentially has an impact on tumor immune microenvironment. Overall, this new discovery is interesting and novel. The hypothesis was in general supported by the experiment evidence.

There are several concerns that need to be addressed to further support the conclusion.

1. In Figure 1 and 2, PD-L1 chromatin association was induced in response to IR. It is interesting that ATRi blocks CHK1 chromatin association but enhances the binding of PD-L1 to chromatin (Fig.1F). Does IR enhance the interaction between ATR and PDL1? Is ATR or ATR kinase activity contributing to the binding of PD-L1 to chromatin?
2. In Figure 2D, Chk1 cytoplasmic fraction was significantly reduced after DNA damage. This data was not consistent with findings in Figure 2C and Figure 1F-H. Please explain this inconsistent data presentation.
3. In early or late responses to IR/DNA damage treatment (Fig. 1F-H and Fig.2C and 2D), association of CHK1 to chromatin was abolished at both conditions. p-p65 reduction/p-TBK1 were examined 1-6 days after NCS or IR in Fig. 2. to indicate NF-kB pathway activation and STING pathway activation. Does PD-L1 knockdown reduce NF-kB/STING pathway activation in early responses to IR/DNA damage 1-6 hrs while association of CHK1 was significantly abolished at these time points? ATRi and CHK1i were used to show a similar phenotype compared to PD-L1 knockdown in NF-kB pathway activation and STING pathway activation. However, key experiment data is lacking to demonstrate that reduced CHK1-association to chromatin in PD-L1-knockdown cells is the underlying cause of reduced NF-kB pathway activation and STING pathway activation. It is possible that ATRi or CHK1i may have indirect inhibitory effect on TBK1 phosphorylation. In these experiments, p-IRF3 might be used to confirm STING pathway activation.
4. In Figure 3E, the authors found the reduced cGAMP level in PD-L1-depleted cells in the presence or absence of NCS treatment. If cells are treated with pure cGAS-STING agonists, does loss of PD-L1 block conventional activation of cGAS-STING pathway given altered cGAMP levels in PD-L1 depleted cells?
5. In Figure 4 and 5, the authors observed PD-L1-depleted cells exhibit a significant reduction of immune-related genes such as IL-6, IL-8, GM-CSF and Ccl-2. PD-L1 rescue-experiments were conducted to demonstrate the importance of restoration of wildtype but not mutant PD-L1 expression. These experiments may show the causative effect of PD-L1 depletion on expression of these genes. However, it is not clear whether it is through association of CHK1 with chromatin association or cGAS/STING pathway. Did these rescuing experiments restore CHK1 chromatin association/p-TBK1 expression?
6. It is unclear what is the biological/cell biology consequence of PD-L1 depletion on these cells in response to DNA damage in addition to molecular changes. Do PD-L1-depleted cells show increased sensitivity to DNA damage (CHK1 pathway alteration)? Do PD-L1-depleted cells alter their tumorigenic features (survival/aggressiveness)?

Referee #3:

This manuscript presents an intriguing exploration of the novel role of PDL1 in regulating the localization of Chk1 and ATR to the nucleus. The authors discuss the binding of PDL1 to Chk1 and ATR and their nuclear/cytoplasmic localization, as well as late NF- κ B phosphorylation, the cGAS/STING connection, and inflammatory cytokine production induced by PDL1. However, several issues reduce the overall enthusiasm for this manuscript. Notably, some controls are missing, and only Western blots are provided without visual proof of the data. Additional cell lines and antibodies could strengthen the conclusions. Overall, while the manuscript addresses a significant topic, the inclusion of additional controls, visual proof, and validation across multiple cell lines would significantly enhance the robustness of the findings.

Major Comments:

1. Figure 1: The authors claim that full-length PDL1 localizes to the nucleus and interacts with ATM/ATR. However, Figure 1A does not prove nuclear localization if whole lysates are used, as indicated in the legend. This interaction might occur in the cytoplasm. Please provide a long exposure of the ATM Western blot in Figure 1A. How do the authors explain the faint band in the Δ CT sample? Co-localization by immunofluorescence (IF) should be shown.
2. Binding Experiment: A binding experiment using a cell line endogenously expressing PDL1 and using a PDL1 antibody can be performed. The obligate binding factor for ATR, ATRIP, should also be investigated. Does ATR binding change the phosphorylation status of PDL1?
3. Figure 1F: Include ATR Western blot in cytoplasmic fractions. Show pChk1 levels for these samples as well.
4. Figure 1H: The authors should also run total lysates to ensure total Chk1 levels are not higher in KO vs. WT cells. In this experiment, pChk1 is also needed. A higher exposure of this Western blot is also necessary. Some Chk1 should be in the nucleus.
5. Cell Line Validation: These experiments should be validated in at least one other cell line to ensure the observed phenotype is not cell line-dependent. The timing of Chk1 phosphorylation in BT549 cells in Extended View Figure 1 does not match the wild-type. If these cells in Extended View Figure 1 are fully knockout, there is a faint band; how was the knockout performed?
6. γ H2AX Signal: The γ H2AX signal in Extended Figure 2 should be monitored over time after DNA damage induction to determine whether PDL1 knockout truly contributes to repair. Persistence of the γ H2AX foci at later time points will show the repair kinetics. Provide some staining images.
7. Figure 2: If ATR/Chk1 is important for the phosphorylation of p65 both in the acute phase and 6 days after treatment, why is the 6-day phosphorylation more affected (Figure 2)? In panel 2E, p65 is observed at day 5. What is the reason for this difference?
8. Figure 3:
 - o The phosphorylation of STING should also be shown in PDL1 WT/KO cells.
 - o Why does it take several days for TBK1 phosphorylation if this is cGAS/STING dependent? Normally, this is more rapid in cells.
 - o Experiments in this figure also need the rescue line.
9. Figure 3E: Baseline cGAMP is lower in knockout cells, which is very strange. This data does not prove that PDL1 is important for cGAMP production after DNA damage. It appears it may be important even without DNA damage.
10. Figure 4: The baseline levels of cytokines differ between WT and KO even without any treatment. Even in control lines, the difference appears meaningful. Please show dot plots with all the data and perform statistical analysis.
11. Figure 4C: The difference in baseline cytokines is again very different between WT and KO, more disparate than the treated cells for WT vs. KO. This should be explained.

Minor Comments:

1. Page 10, Line 6: Please confirm the reference.
2. Cell Viability: The viability of the cells in WT/KO under treatment conditions should be shown. This could explain reduced transcription/translation and not specific effects.
3. PDL1 Antibody: Another PDL1 antibody should be used to validate one of the Western blots.
4. HDAC2: HDAC2 was not extensively studied and feels disconnected from the rest of the manuscript.

- Point-by-point response to reviewers' questions -

We would like to thank the editor for recognizing the significance of our work and organizing the constructive comments and suggestion from the three reviewers, which have been very helpful in guiding us to further improve our manuscript during this round of revision. Below we answer to the questions/criticisms raised by the referees.

Referee #1:

The submitted manuscript addresses the question to what extent PD-1/PD-L1 has a direct effect on the regulation of the DNA damage response after genotoxic stress. The authors observed that nuclear PD-L1 activates the ATR-Chk1 signaling pathway upon genotoxic stress and regulates the expression of proinflammatory chemokines. They observed that PD-L1 interacts with ATR and is essential for Chk1 activation and chromatin binding. The cGAS-STING and NF- κ B activation in the late phase of DNA damage response was inhibited by PD-L1 deletion or by inhibitors of ATR and Chk1. Consequently, induction of proinflammatory chemocytokines at this phase was inhibited by deletion of PD-L1 or its mutation, which inhibits nuclear localization, or by the HDAC2 inhibitor santacruzamate A, which blocks PD-L1 deacetylation and nuclear localization. Conversely, chemocytokine induction was enhanced by the p300 inhibitor C646, which accelerated PD-L1 nuclear localization.

These results suggest that nuclear PD-L1 can directly enhance DNA damage repair and thus could also influence the tumor's intracellular response to immune checkpoint inhibitors.

The manuscript deals with a biologically important mechanism whose elucidation could be of great importance for tumor therapy. I therefore consider the manuscript to be very relevant and interesting. However, I do not consider it to be sufficiently mature in its current form. I would therefore like to make a few suggestions for further improvement of the manuscript before I could agree to its publication in EMBO reports.

[Response]

We appreciate the reviewer for acknowledging our efforts. We also thank the reviewer for recognizing the novelty and the significant impact of this study. More importantly, we sincerely thank the reviewer for raising the constructive comments to help us further strengthen our manuscript. Below please find the point-to-point response to the reviewer's critiques.

As the authors themselves note in their discussion, the most direct evidence of the effect of PD-L1 on the phosphorylation of CHK1 would be on the initiation of the cell cycle checkpoint, the activation of dormant replication origins, the slowing down of replication, the accumulation of RPA foci, to name just a few very important mechanisms. Since the further experiments are based on this hypothesis, I consider it necessary that at least two of these processes are demonstrated by the authors.

[Response]

We thank the reviewer for the suggestion. First, we examined the effect of PD-L1 on cell cycle checkpoint using EdU incorporation to measure replicative DNA synthesis (Revised Figure EV2A). In WT MDA-MB-231 cells, EdU incorporation significantly decreased after IR exposure, indicating cell cycle arrest due to the intra-S phase checkpoint. In contrast, PD-L1 KO cells failed to suppress the DNA synthesis after IR irradiation, suggesting that Chk1 does not correctly function in these cells.

Next, we compared DNA damage-induced RPA2 foci formation between WT and PD-L1 KO MDA-MB-231 cells. However, no significant difference in RPA2 foci formation was observed between the cell lines. This is likely because RPA2 foci formation is not downstream of ATR or Chk1. We therefore tested whether the phosphorylation of RPA2 at Ser33, catalyzed by ATR, and phosphorylation of Cdc25, well known substrate of Chk1, were reduced in PD-L1 KO cells. Both substrates phosphorylated after IR (Revised Figure 1G) and NCS (Revised Figure H) were remarkably suppressed by PD-L1 depletion.

Together, these findings support the dysregulation of the cell cycle checkpoint in PD-L1 KO cells.

Furthermore, the functions of ATR and ATM cannot be strictly separated from each other. It would therefore make sense to carry out at least one of the experiments with an ATM inhibitor for comparison.

[Response]

As kindly suggested by the reviewer, we examined the DNA damage-induced phosphorylation of TBK1 using the specific ATM inhibitor, KU-55933 (Revised Figure EV3D). Although it was mildly suppressed, DNA damage-induced phosphorylation of both TBK1 and Chk1 was observed in cells treated with KU-55933, despite a reduction

in ATM autophosphorylation, which indicates that the inhibitor is effective. Based on the results, ATR, rather than ATM, is involved in the activation of DNA damage-induced cGAS-STING signaling.

Revised Figure EV3D

It is not clear why etoposide was used for the induction of DNA damage. The data shown so far are based on the DNA-damaging effect of irradiation, Neocarzinostatin and Doxorubicin. Would it therefore make more sense to investigate their effect on the formation of DNA damage? Or, conversely, also show the effect of etoposide treatment on CHK1 activation, chromatin binding and proliferation activity?

[Response]

We thank the reviewer for raising this concern. In this revision, we have re-examined the analysis using NCS. Another reviewer also suggested demonstrating γ H2AX through IF analysis. Therefore, we have monitored γ H2AX-positive cells following DNA damage by IF analysis (Revised Figure EV2A). In WT MDA-MB-231 cells, γ H2AX foci triggered by NCS gradually decreased. However, in PD-L1 KO cells, foci formation persisted even after 24 hours, suggesting an impaired DNA damage response due to PD-L1 depletion.

Revised Figure. EV2A

The authors extensively describe the importance of CHK1 for the formation of RAD51 and the recruitment of BRCA1, but show no data on whether this is impaired in their system. This would be very interesting, especially in comparison to the HA-PD-L1 deletion mutant.

[Response]

As kindly requested by the reviewer, we have tried to investigate the DSB-induced foci formation of BRCA1 and Rad51. However, immunostaining for BRCA1 was unsuccessful because no effective antibody was currently available in our laboratory. We therefore tested this possibility by examining BARD1, which is an essential binding partner of BRCA1. In the previous report we cited (Kornepati *et al*, 2022), BARD1 interacted with PD-L1 and this interaction was critical for BRCA1 retention at DSBs. However, IR- or NCS-induced BARD1 foci were not affected by PD-L1 depletion (Figure R1), nor was RAD51 foci formation. Thus, the observed failure of DSB repair may not be due to direct effects of PD-L1, ATR, or Chk1 on homologous recombination factors, but rather may be caused by an indirect effect of ATR or Chk1, for example by cell-cycle arrest defect. Based on these results, and because this is not the main focus of this paper, we have removed the description of RAD51 (page 10 line 15) and BRCA1 (page 11, line 11) from the text.

Figure R1

I also do not consider the exclusive use of the CellTiterGlo assay to describe the effects on cell survival to be sufficient. The data obtained in this way only indicate that the cells were inhibited in proliferation for 48 hours, but do not provide any information about cell death. At least a colony formation test or an adequate test to determine survival after a longer period of time would be necessary here.

[Response]

As kindly instructed by the reviewer, we have performed the colony formation assay (Revised Figure EV2B). The results were consistent with previous findings. We appreciate the suggestion to improve the quality of the manuscript.

Revised Figure EV2B

Minor points:

It would be useful to include data from different experiments in the main manuscript, such as DNA damage or survival. The manuscript consists primarily of Western blots and mRNA expression data. This is very difficult to read and understand. For protein expression, graphs with quantifications would also be useful for easier comparability.

[Response]

We apologize for the difficulty. The data indeed consisted primarily of Western blots and mRNA expression. In the revised version, immunofluorescence data of IR-induced PD-L1-ATR colocalization was added (Revised Figure 1D, E and F) in the main manuscript. We also considered whether to transfer data on replication arrest (Revised Figure EV1D), DSB repair (Revised Figure EV2A) and clonogenic survival (Revised Figure EV2B) to the main manuscript. However, the space was not enough because of the limitation of the journal. I would appreciate it if the reviewer could accept this situation.

For quantification of protein expression, the most important point in revised Figure. 2, Chk1 phosphorylation, is shown in graphs derived from $n = 2$ or 3 biological

replicates (Revised Figure 2E, EV1A and B). All other data were also repeated at least twice, and the number of biological and technical replicates is clearly stated in each Figure legend or Methods.

Revised Figure 2E

Revised Figure EV1A

Revised Figure EV1 B

The authors describe that for the CellTiterGlo assay 800 cells were seeded in 96-well plates; it would be more interesting to know how many cells were seeded per well.

[Response]

We sincerely apologize for the previous incorrect description of the cell viability assay. The correct procedure involved seeding 800 cells per well in a 96-well plate. This data has been removed as it has been replaced by colony formation assay data.

Figure 1B: The irradiation dose should be indicated either in the Figure or its legend. And wouldn't at least one more dose point make sense, since at 10 Gy a relatively large amount of cells should already be dead?

[Response]

We apologize for the insufficient explanation of the experimental conditions in previous Figure 1B. In the experiment shown in the previous Figure 1B (revised Figure 2A), the cells were exposed to 10 Gy of X-ray irradiation. We have added the information in revised Figure legends, on page 33 line 1. We used 10 Gy IR in experiments with short follow-up (up to 4 hours) and 2 Gy IR in experiments with long follow-up (up to 6 days). The dosages were carefully determined to avoid affecting cell viability. Because such concerns may arise from readers, we have added Figures of phase contrast data for cultured cells after exposure to 10 Gy IR (Revised Figure EV1C) and 2 Gy IR (Revised

Figure EV3A) and before harvesting. No changes in morphology or survival were observed. Additionally, we conducted a similar experiment using 5 Gy IR exposure (Figure R2). The phosphorylation levels of CHK1 were also reduced in PD-L1 KO cells with this IR dose.

Revised Figure EV1C

Figure R2

WT and PD-L1 KO MDA-MB-231 cells were exposed to 5 Gy IR and incubated for the indicated hours. Cell lysates were analyzed by Western blotting.

Revised Figure EV3A

Figure 2A: p-p65 S536? Is this a control? And wouldn't it make more sense to show this at the bottom of the Figure so that the important data can be seen immediately?

[Response]

Based on the report that ATM/ATR activates NF- κ B in response to DNA damage (Kang et al., *Science* (2015)), the aim of our experiments in Figure 2A (Revised Figure 3A) was to investigate whether phosphorylation of p65, a subunit of NF- κ B, is triggered by DNA damage in ATR dependent manner. In parallel, as a positive control for ATR inhibitor VE-821, the phosphorylation blot of Chk1 which is a direct target of ATR was included in Figure 2A. Therefore, the p65 phosphorylation is shown in the top panel.

Referee #2:

The authors of this manuscript reported a new function of nuclear PD-L1 in regulating DNA damage response pathway the ATR-Chk1 signaling and inducing proinflammatory chemo cytokines genotoxic stress. PD-L1 is found to interact with ATR through a proteomic analysis and consequently required for Chk1 activation. It is interesting that PD-L1 depletion inhibits DNA damage -triggered cGAS-STING and NF-kappaB activation in a similar manner as inhibition of ATR and Chk1 activation. Furthermore, experiment data showed that altered PD-L1 localization in nuclear by the HDAC2 inhibitor (reduced localization) or the p300 inhibitor (enhanced localization) changed the induction of proinflammatory chemocytokines in the late phase of DNA damage response. Collectively, these findings uncovered a new role of nuclear PD-L1 in regulating DNA damage-triggered immune responses, which potentially has an impact on tumor immune microenvironment. Overall, this new discovery is interesting and novel. The hypothesis was in general supported by the experiment evidence. There are several concerns that need to be addressed to further support the conclusion.

[Response]

We thank the reviewer for recognizing the novelty and potentially wide interest for both basic and translational cancer research field of our study. We also thank the reviewer for sharing with us the insightful analyses and raising constructive comments to help us further strengthen our manuscript. Below, please find the point-to-point response to the reviewer's concerns.

1. In Figure 1 and 2, PD-L1 chromatin association was induced in response to IR. It is interesting that ATRi blocks CHK1 chromatin association but enhances the binding of PD-L1 to chromatin (Figure1F). Does IR enhance the interaction between ATR and PDL1? Is ATR or ATR kinase activity contributing to the binding of PD-L1 to chromatin?

[Response]

We thank the reviewer for the excellent suggestion. We confirmed increased chromatin recruitment of PD-L1 after IR exposure in another cell line, BT-549 (Figure R3). However, we did not observe inhibition of increased chromatin recruitment of PD-L1 by ATR inhibitor in BT-549 cells, suggesting that this effect of the ATR inhibitor is cell line dependent. Therefore, we did not discuss this point in the current manuscript. However, this is definitely an important point and is worth following up in the future.

Figure R3

Western blot analysis of chromosomal and cytoplasmic fractions from WT BT-549 cells, either untreated or treated with 1 μ M ATR inhibitor VE-821, followed by exposure to 2 Gy IR and incubation for the indicated hours.

2. In Figure 2D, Chk1 cytoplasmic fraction was significantly reduced after DNA damage. This data was not consistent with findings in Figure 2C and Figure 1F-H. Please explain this inconsistent data presentation.

[Response]

We thank the reviewer for highlighting this important concern. The different expression level could be due to the DNA damaging agent; NCS in Figure 2C (revised Figure 3C) and IR in Figure 2D (revised Figure 3D), and different timing after DNA damage; up to 2 hours in Figure 1F-H (revised Figure 2F-H) and 4-5 days in Figure 2C (revised Figure 3C) and 6 days in Figure 2D (revised Figure 3D). To clarify this discrepancy, we compared cytoplasmic Chk1 levels from days 4 to 6 after 2 Gy IR (Figure R4). Chk1 expression in the cytoplasmic fraction was further decreased on day 6 post-IR compared with days 4 and 5 post-IR. We also repeated the experiment shown in the previous Figure 2D (Figure R5) as well as all western blot data at least twice to confirm the result.

**Figure R4**

Western blotting analysis of chromosomal and cytoplasmic fractions from WT or PD-L1 KO MDA-MB-231 cells exposed to 2 Gy IR and incubated for the indicated hours.

**Figure R5**

3. In early or late responses to IR/DNA damage treatment (Figure. 1F-H and Figure.2C and 2D), association of CHK1 to chromatin was abolished at both conditions. p-p65 reduction/p-TBK1 were examined 1-6 days after NCS or IR in Figure 2. to indicate NF-kB pathway activation and STING pathway activation. Does PD-L1 knockdown reduce NF-kB/STING pathway activation in early responses to IR/DNA damage 1-6 hrs while association of CHK1 was significantly abolished at these time points?

[Response]

We thank the reviewer for raising this concern. Although PD-L1 promotes activation of ATR and Chk1 at both early and late phases after DNA damage, we speculate that PD-L1- and ATR/Chk1-dependent STING activation and subsequent NF-kB activation only occur at late phase, since cytoplasmic DNA is triggered by micronuclear rupture after cell-cycle progression.

ATRⁱ and CHK1ⁱ were used to show a similar phenotype compared to PD-L1 knockdown in NF-kB pathway activation and STING pathway activation. However, key experiment data is lacking to demonstrate that reduced CHK1-association to chromatin in PD-L1-knockdown cells is the underlying cause of reduced NF-kB pathway activation and STING pathway activation. It is possible that ATRⁱ or CHK1ⁱ may have indirect inhibitory effect on TBK1 phosphorylation.

[Response]

We fully agree with the reviewer's insightful comments that the phenotypes of WT cells treated with ATRⁱ or CHK1ⁱ could not exclude an indirect effect of PD-L1, which

prompted us to conduct further analysis under experimental conditions with enforced ATR activity in KO cells. We compared the phosphorylation status of TBK1 in PD-L1 KO cells with the ATR activator Garcinone C. DNA damage-induced TBK1 phosphorylation abolished in PD-L1 KO cells was restored by Garcinone C treatment (revised Figure 4E), suggesting that PD-L1-dependent activation of the cGAS-STING pathway requires ATR activity. We appreciate the reviewers' comment that significantly improved the quality of the manuscript.

revised Figure 4E

In these experiments, p-IRF3 might be used to confirm STING pathway activation.

[Response]

Unfortunately, we were unable to detect IRF3 phosphorylation in MDA-MB-231 cells. However, we observed DNA damage-induced phosphorylation of IRF3 in BT-549 cells accompanied by phosphorylation of TBK1 and STING, that all inhibited in PD-L1 KO cells but restored by HA-PD-L1 add-back (revised Figure EV3C). We speculate that whether the downstream of the cGAS-STING pathway is primarily NF- κ B or IRF may vary depending on the cell line. In MDA-MB-231 cells, cGAS-STING primarily activates the TBK1/NF- κ B pathway to induce inflammation, whereas in BT-549 cells, it activates both the TBK1/NF- κ B and canonical TBK1/STING/IRF3 signaling pathways. In either case, PD-L1 is critical for NF- κ B-induced inflammation via cGAS-STING.

revised Figure EV3C

4. In Figure 3E, the authors found the reduced cGAMP level in PD-L1-depleted cells in the presence or absence of NCS treatment. If cells are treated with pure cGAS-STING agonists, does loss of PD-L1 block conventional activation of cGAS-STING pathway given altered cGAMP levels in PD-L1 depleted cells?

[Response]

We thank the reviewer for this excellent suggestion. Following the recommendation, we performed an ELISA assay using synthetic dsDNA (dAdT) as a source of cytoplasmic DNA to induce the cGAS-STING pathway (revised Figure EV3F). cGAMP production after transfection with dAdT was not suppressed by PD-L1 deletion, suggesting that PD-L1 functions upstream of cGAS or cytoplasmic DNA generation.

In the revised manuscript, we also performed the cGAMP ELISA assay using the ATR activator Garcinone C to test whether it rescue PD-L1 deletion (revised Figure 4F, replacing the previous Figure 3E). Notably, NCS-induced cGAMP production, which was suppressed in PD-L1 KO cells, was restored by garcinone C, indicating that the cGAS-STING pathway was intact in PD-L1 KO cells in the presence of ATR activity. These data indicate that PD-L1 is involved upstream of cGAS, including the step of cytoplasmic DNA generation after DNA damage via ATR activation.

revised Figure EV3F

revised Figure 4F

5. In Figure 4 and 5, the authors observed PD-L1-depleted cells exhibit a significant reduction of immune-related genes such as IL-6, IL-8, GM-CSF and Ccl-2. PD-L1 rescue-experiments were conducted to demonstrate the importance of restoration of wildtype but not mutant PD-L1 expression. These experiments may show the causative effect of PD-L1 depletion on expression of these genes. However, it is not clear whether it is through association of CHK1 with chromatin association or cGAS/STING pathway. Did these rescuing experiments restore CHK1 chromatin association/p-TBK1 expression?

[Response]

We thank the reviewer for pointing out this important concern. To answer the reviewer's question, we performed rescue experiments on cGAS-STING reactivation. NCS-induced

phosphorylation of TBK1, suppressed in PD-L1 KO cells, was restored by exogenous HA-PD-L1 expression in MDA-MB-231 cells (revised Figure 4A) and BT-549 cells (revised Figure EV3C, shown in response to comment #3 above), as expected.

revised Figure 3C

Regarding Chk1 chromatin binding, the ATR activator Garcinone C, which restored suppressed cGAMP production (revised Fig. 4F) and expression of IL-6 and GM-CSF (revised Fig. 5C) in PD-L1 KO cells, also restored Chk1 binding to chromatin (revised Figure EV3E).

revised Figure EV3E

6. It is unclear what is the biological/cell biology consequence of PD-L1 depletion on these cells in response to DNA damage in addition to molecular changes. Do PD-L1-depleted cells show increased sensitivity to DNA damage (CHK1 pathway activation)? Do PD-L1-depleted cells alter their tumorigenic features (survival/aggressiveness)?

[Response]

We appreciate the reviewer's comment. To answer this question, we performed the colony formation assay (Revised Figure EV2B). To answer this question, we performed colony formation assays (revised Fig. EV2B). PD-L1 deletion sensitized cells to the DNA damaging agent doxorubicin in addition to reducing inflammation, which is important *in vivo*.

Revised Figure EV2B

Referee #3:

This manuscript presents an intriguing exploration of the novel role of PDL1 in regulating the localization of Chk1 and ATR to the nucleus. The authors discuss the binding of PDL1 to Chk1 and ATR and their nuclear/cytoplasmic localization, as well as late NF- κ B phosphorylation, the cGAS/STING connection, and inflammatory cytokine production induced by PDL1. However, several issues reduce the overall enthusiasm for this manuscript. Notably, some controls are missing, and only Western blots are provided without visual proof of the data. Additional cell lines and antibodies could strengthen the conclusions. Overall, while the manuscript addresses a significant topic, the inclusion of additional controls, visual proof, and validation across multiple cell lines would significantly enhance the robustness of the findings.

[Response]

We appreciate the reviewers for recognizing the novelty and significant impact of this work. We would also like to thank the reviewers for sharing their insightful analyses and providing constructive comments that helped us further strengthen the manuscript. Below, please find the point-to-point response to the reviewer's concerns.

Major Comments:

1. Figure 1: The authors claim that full-length PDL1 localizes to the nucleus and interacts with ATM/ATR. However, Figure 1A does not prove nuclear localization if whole lysates are used, as indicated in the legend. This interaction might occur in the cytoplasm. Please provide a long exposure of the ATM Western blot in Figure 1A. How do the authors explain the faint band in the Δ CT sample? Co-localization by immunofluorescence (IF) should be shown.

[Response]

We appreciate the important comment. We attempted to indirectly demonstrate that PD-L1 binds to ATR in the nucleus by using Δ CT mutant that we previously showed cannot translocate intracellularly or into the nucleus (Gao *et al.*, *Nat Cell Biol* 22: 1064-1075, 2020). As the reviewer pointed out, the binding shown in Figure 1A is in the total eluate, and the data itself does not directly demonstrate binding in the nucleus. To compensate this issue, we performed immunofluorescence staining experiments to examine whether PD-L1 actually colocalizes with ATR in the nucleus as suggested by the reviewer. As suitable PD-L1 antibodies for immunofluorescence were not available, the PD-L1 knockout (KO) TNBC cell line MDA-MB-231 with stable expression of HA-tagged PD-L1 was used (revised Figure 1C). Immunostaining with anti-HA antibodies showed that PD-L1 was predominantly localised in the cytoplasm and plasma membrane in formalin-fixed cells without pre-extraction (revised Figure 1D). However, pre-extraction treatment of cells revealed that PD-L1 also forms nuclear foci that partially co-localise with ATR (revised Figure 1E), which increase after IR exposure (revised Figure 1F). The specificity of the antibody was validated by PD-L1 KO cells expressing an empty vector (revised Figure 1D, E).

revised Figure 1C

revised Figure 1D

revised Figure 1E

revised Figure 1F

U2OS fibroblast
(The Human Protein ATLAS)

The Human Protein ATLAS (<https://www.proteinatlas.org/ENSG00000120217-CD274/subcellular>) exhibits nuclear PD-L1 in U2OS cells and fibroblasts as shown above.

Regarding to the faint bands in the Δ CT sample, they were indeed visible with longer exposures (Fig. R6). We speculated that it was a nonspecific precipitation by the antibody. We therefore repeated the experiment and after careful washing of the immunoprecipitates, this faint band was no longer detectable (revised Figure 1A).

Figure R6

Previous Figure 1A with the addition of a long exposure of the ATM blot.

revised Figure 1A

2. Binding Experiment: A binding experiment using a cell line endogenously expressing PDL1 and using a PDL1 antibody can be performed. The obligate binding factor for ATR, ATRIP, should also be investigated. Does ATR binding change the phosphorylation status of PDL1?

[Response]

We thank the reviewer for this excellent suggestion. Because no effective antibody to precipitate endogenous ATR nor PD-L1 was currently available in our laboratory, we used anti-ATRIP antibody. The antibody co-immunoprecipitated endogenous ATR and PD-L1 as expected (revised Figure 1B). Given that PD-L1 does not have an ATR-phosphorylated SQ/TQ motif, we speculate that PD-L1 is not a substrate for ATR.

revised Figure 1B

3. Figure 1F: Include ATR Western blot in cytoplasmic fractions. Show pChk1 levels for these samples as well.

[Response]

As requested by the reviewer, the cytoplasmic fractions were subjected to immunoblot analysis with an ATR antibody. In addition, pChk1 blots from both the chromosomal and cytoplasmic fractions were included in revised Figure 2F (previous Figure 1F). The phosphorylation levels of Chk1 in both the chromosomal and cytoplasmic fractions increased after DNA damage were inhibited by the ATR inhibitor VE-821.

revised Figure 2F

4. Figure 1H: The authors should also run total lysates to ensure total Chk1 levels are not higher in KO vs. WT cells. In this experiment, pChk1 is also needed. A higher exposure of this Western blot is also necessary. Some Chk1 should be in the nucleus.

[Response]

As requested by the reviewer, we have added the blots of the total lysate. In addition, we re-run the chromatin fraction samples for Chk1 blotting and showed longer exposure (Revised Figure 1H). Chk1 was barely detectable in the chromatin fraction of PD-L1 KO cells.

Revised Figure 2H

5. Cell Line Validation: These experiments should be validated in at least one other cell line to ensure the observed phenotype is not cell line-dependent. The timing of Chk1 phosphorylation in BT549 cells in Extended View Figure 1 does not match the wild-type. If these cells in Extended View Figure 1 are fully knockout, there is a faint band; how was the knockout performed?

[Response]

The KO cells were kindly provided by Dr. Mien-Chie Hung and were generated using CRISPR/Cas9 KO plasmids from Santa Cruz Biotechnology (Jiao et al., Clinical Cancer Research, 2017). The faint band detected in Extended View Figure 1A and B could be non-specific products. We confirmed the knockout efficiency by western blotting using two commercial antibodies (Figure R7).

Figure R7

To make the phosphorylated Chk1 data in Figure EV1A and B more clear, we presented the data as graphs derived from $n = 2$ biological replicates. DNA damage-triggered Chk1 phosphorylation was attenuated in PD-L1 KO cells compared to WT BT-549 cells, although the timing of phosphorylation varied depending on the stimulus (Revised Figure EV1C and EV1D).

Revised Figure EV1A

Revised Figure EV1 B

6. γ H2AX Signal: The γ H2AX signal in Extended Figure 2 should be monitored over time after DNA damage induction to determine whether PDL1 knockout truly contributes to repair. Persistence of the γ H2AX foci at later time points will show the repair kinetics. Provide some staining images.

[Response]

As kind instruction by the reviewer, we have monitored the recovery of DNA damage-induced γ H2AX nuclear foci by IF analysis (Revised Figure EV2A). Although the foci steadily declined in WT MDA-MB-231 cells, they remained at significantly higher levels in PD-L1 KO cells even after 24 h (Fig. EV2A).

Revised Figure. EV2A

7. Figure 2: If ATR/Chk1 is important for the phosphorylation of p65 both in the acute phase and 6 days after treatment, why is the 6-day phosphorylation more affected (Figure 2)? In panel 2E, p65 is observed at day 5. What is the reason for this difference?

[Response]

We thank the reviewer for raising this concern. Although PD-L1 promotes activation of ATR and Chk1 at both early and late phases after DNA damage, we speculate that PD-L1- and ATR/Chk1-dependent STING activation and subsequent NF- κ B activation (phosphorylation of p65) only occur at late phase, since cytoplasmic DNA is triggered by micronuclear rupture after cell-cycle progression. NF- κ B activation in the acute phase appears to be less dependent on ATR-Chk1. Other groups have shown that micronuclei formation was observed 3-6 days after IR irradiation in MCF10A cells (Harding et al., Nature, 2017) or 5-8 days after etoposide treatment in IMR90 cells (Dou et al., Nature, 2017). The difference in time between these several days appears to depend on the rate of cell-cycle progression following DNA damage.

8. Figure 3: (revised Figure 4)

o The phosphorylation of STING should also be shown in PDL1 WT/KO cells.

[Response]

As kindly requested by the reviewer, we examined the phosphorylation status of STING. Unfortunately, we were unable to detect STING phosphorylation in MDA-MB-231 cells. However, we observed DNA damage-induced phosphorylation of STING in BT-549 cells accompanied by phosphorylation of TBK1 and IRF3, that all inhibited in PD-L1 KO cells but restored by HA-PD-L1 add-back (revised Figure EV3C). We speculate that whether the downstream of the cGAS-STING pathway is primarily NF- κ B or IRF may vary depending on the cell line. In MDA-MB-231 cells, cGAS-STING primarily activates the TBK1/NF- κ B pathway to induce inflammation, whereas in BT-549 cells, it activates both the TBK1/NF- κ B and canonical TBK1/STING/IRF3 signaling pathways. In either case, PD-L1 is critical for NF- κ B-induced inflammation via cGAS-STING.

revised Figure EV3C

o Why does it take several days for TBK1 phosphorylation if this is cGAS/STING dependent? Normally, this is more rapid in cells.

[Response]

We thank the reviewer for raising this concern. Whereas some reports demonstrated cGAS-STING activation in earlier time points, that at late phase is common, since cytoplasmic DNA is triggered by micronuclear rupture after cell-cycle progression. For example, it has been shown that cGAS-STING activation caused by micronuclei formation was observed 3-6 days after IR irradiation in MCF10A cells (Harding et al., Nature, 2017) or 5-8 days after etoposide treatment in IMR90 cells (Dou et al., Nature, 2017).

o Experiments in this Figure also need the rescue line.

[Response]

We thank the reviewer for pointing out this important concern. In response to the suggestion, we performed rescue experiments on cGAS-STING reactivation. NCS-induced phosphorylation of TBK1, suppressed in PD-L1 KO cells, was restored by exogenous HA-PD-L1 expression in MDA-MB-231 cells (revised Figure 4A) and BT-549 cells (revised Figure EV3C, shown in response to the first question of comment #8 above).

revised Figure 4A

9. Figure 3E: Baseline cGAMP is lower in knockout cells, which is very strange. This data does not prove that PDL1 is important for cGAMP production after DNA damage. It appears it may be important even without DNA damage.

[Response]

We thank the reviewer for raising this important concern. In cancer cells with high genomic instability due to dysfunctional DNA damage response, replication stress leads to mis-segregation of chromosome fragments. Consequently, cytoplasmic DNA may accumulate in MDA-MB-231 cells, subsequently activating cGAS and resulting in cGAMP production. We added sentences “Baseline cGAMP level in MDA-MB-231 cells, which were relatively higher than in BT549 cells (Fig. 4F and EV3F), was also inhibited

by PD-L1 KO. This could be due to higher genomic instability in this cell line.” in page 14, line 16 to page 15, line 1.

10. Figure 4: The baseline levels of cytokines differ between WT and KO even without any treatment. Even in control lines, the difference appears meaningful. Please show dot plots with all the data and perform statistical analysis.

[Response]

As requested by the reviewer, we have changed the qRT-PCR results from bar graphs to dot plots.

The baseline levels of cytokines suppressed by PD-L1 is likely due to the same reason as explained above for cGAMP. We added sentences “Similar to the observation made for cGAMP, the baseline levels of chemocytokines in MDA-MB-231 cells were also suppressed by PD-L1 KO (Fig. 5, EV4A and B). This suggests that PD-L1 mediates inflammation via endogenous DNA damage in addition to that caused by exogenous DNA damage.” in page 16, line 4 to 7.

11. Figure 4C: The difference in baseline cytokines is again very different between WT and KO, more disparate than the treated cells for WT vs. KO. This should be explained.

[Response]

We thank the reviewer for raising this concern. The baseline levels of secreted cytokines suppressed by PD-L1 is likely due to the same reason as discussed above. The discrepancy in fold changes between qRT-PCR and ELISA results for cytokines, with and without DNA damage, may be due to differences in the sensitivity of each assay.

Minor Comments:

1. Page 10, Line 6: Please confirm the reference.

[Response]

We apologize our fault for the citation. We have deleted it in the revised manuscript.

2. Cell Viability: The viability of the cells in WT/KO under treatment conditions should be shown. This could explain reduced transcription/translation and not specific effects.

[Response]

The dosages of DNA damaging agents were carefully determined to avoid affecting cell viability. According to the reviewer’s suggestion, we have added Figures of phase contrast data for cultured cells after the DNA damages and before harvesting (Revised Figure EV1C and EV3A). No changes in morphology or survival were observed.

Revised Figure EV1C

Revised Figure EV3A

3. PDL1 Antibody: Another PDL1 antibody should be used to validate one of the Western blots.

[Response]

As requested by the reviewer, we performed the immunoblotting analysis using another commercial PD-L1 antibody (GeneTex) and confirmed the PD-L1 knockout (Revised Figure 1B and Figure R7 shown above).

Revised Figure 1B

4. HDAC2: HDAC2 was not extensively studied and feels disconnected from the rest of the manuscript.

[Response]

We thank the reviewer for raising this concern, but we did not remove the data from the manuscript. The reason is that the deacetylation of PD-L1 by HDAC2 and the resulting inhibition of PD-L1 translocation into the nucleus have been analyzed in detail in a previous paper (Gao *et al.*, *Nat Cell Biol* 22: 1064-1075, 2020), and the data in the current paper is based on that. In this paper, we analyzed PD-L1 mutants at the HDAC2 site of action, HDAC inhibitors, and p300 inhibitors that have the opposite effect to HDAC2. We believe this data is very important as it may form the basis for the clinical application of these drugs. With this in mind, we have added some content from the previous paper to lines 1 to 2 on page 18 of the text.

Additional changes:

1. As a new Figure 1 has been added, all previous figures have been renumbered accordingly.
2. Some changes have been made to the abstract due to additional data.
3. The previous integrated 'Results and Discussion' section has been separated into 'Results' and 'Discussion' sections.
4. All the data are derived from at least $n = 2$ biological replicates. This is described at the end of each figure legend section and in some of the methods.
5. Three additional references were added for additional methods.
6. Additional methods for establishment of PD-L1 add-back cell lines, anti-ATRIP immunoprecipitation, immunofluorescence microscopy, clonogenic survival assay, and purification of chromatin fraction were added in Method section.
7. Page 33, line 15: We found an unrelated sentence "(H) Western blotting analysis of chromosomal and cytoplasmic fractions from WT or PD-L1 KO MDA-MB-231 cells treated with 100 ng/ml NCS." in the previous version and deleted it in the revised version.
8. Details of all changes are highlighted in red in the attached "Highlighted Text" file.

Dear Prof. Ohta,

Thank you for the submission of your revised manuscript to our editorial offices. I have now received the reports from two of the three referees that I asked to re-evaluate the study, you will find below. As you will see, referees #1 and #3 now fully support publication of the study in EMBO reports. Referee #2 remained completely unresponsive to my invitations to re-assess the manuscript. Going through your p-b-p-response myself, I consider the comments however as adequately addressed.

Before I can proceed with formal acceptance, I have these editorial requests I ask you to address in a final revised manuscript:

- Please provide a more active title. How about:

Nuclear PD-L1 triggers tumour-associated inflammation upon DNA damage

- Please provide the abstract written in present tense throughout.

- Please make sure that all figure panels (main, EV and Appendix figures) are called out separately and sequentially. There is a panel 5D called out (page 16) but there is no panel D in the figure file nor is it mentioned in the legend. Please check. There is a 'Supplementary Table' called out twice but no such table is uploaded. Or do you refer to the reagents and tools table? If yes, please change the callout (see Reagents and Tools Table). Please check.

- The nomenclature for the EV figure legends and the individual figure files is not correct. It should be "Figure EVx", not "Expanded View Figure x". Please fix this.

- Please add scale bars of similar style and thickness to all microscopic images (presently shown only in the Appendix, it seems), using clearly visible black or white bars (depending on the background). Please place these in the lower right corner of the images themselves. Please do not write on or near the bars in the image but define the size in the respective figure legend. Presently, some scale bars are rather thin and hard to see.

- Please check that the number "n" for how many independent experiments were performed, their nature (biological versus technical replicates), the bars and error bars (e.g. SEM, SD) and the test used to calculate p-values is indicated in the respective figure legends (main and EV figures). Please also check that all the p-values are explained in the legend, and that these fit to those shown in the figure. Please provide statistical testing where applicable. Please avoid the phrase 'independent experiment', but clearly state if these were biological or technical replicates. Please also indicate (e.g. with n.s.) if testing was performed, but the differences are not significant. In case n=2, please show the data as separate datapoints without error bars and statistics.

See also:

<http://www.embopress.org/page/journal/14693178/authorguide#statisticalanalysis>

If n<5, please show single datapoints for diagrams. Moreover:

- Please note that the error bars are not defined in the legends of figures 1F, 2E, 4F, 5A-C; 6B, D, E, F; EV3 F; EV4 A-C; EV5 B, D.

- Please note that the measure of center for the error bars needs to be defined in the legends of figures EV 1 D.

- Please note that the legends for EV 5 is not provided in the sequential manner (legend for sub-figure D is provided before legend of figure C). This needs to be rectified.

- Please indicate what ** represents; if this represents p value(s), please indicate the statistical test used and where appropriate and the exact p value in the legend of figure 5C.

- Please note that the exact p values are not provided in the legends of figures 1F, 4F, 5A-B; 6B, D, E, F; EV1 D, EV2 A, EV4 A, B; EV5 B, D.

- Please indicate the statistical test used for data analysis in the legends of figures 1F, 4F, 5A-B; 6B, D, E, F; EV1 D, EV2 A; EV3 F; EV4 A-C; EV5 B, D."

- Please add to each legend (main, EV figures, where applicable) a 'Data Information' section explaining the statistics used or providing information regarding replicates and scales. See:

- Please remove the instructions from the reagents and tools table.

- Thank you for providing the requested source data. Please upload this as one folder per figure (with all files for one figure in one folder and ZIPed together) and one folder for all the source data for the EV figures.

In addition, I would need from you uploaded separately:

- a short, two-sentence summary of the manuscript (not more than 35 words).

- two to four short (!) bullet points highlighting the key findings of your study (two lines each).

- a schematic summary figure as separate file that provides a sketch of the major findings (not a data image) in jpeg or tiff format

(with the exact width of 550 pixels and a height of not more than 400 pixels) that can be used as a visual synopsis on our website.

Best,

Referee #1:

I would like to thank the authors for accepting all my and the suggestions of the other reviewers for further improvement of the manuscript and for validating them with appropriate experiments.

The authors have done an excellent job of addressing all questions and I think the manuscript is now suitable for publication.

Referee #3:

The manuscript is suitable for publication in EMBO reports without further revisions.

All editorial and formatting issues were resolved by the authors.

Prof. Tomohiko Ohta
St. Marianna University Graduate School of Medicine
Department of Translational Oncology
2-16-1, Sugao, Miyamae-ku
Kawasaki 216-8511
Japan

Dear Prof. Ohta,

I am very pleased to accept your manuscript for publication in the next available issue of EMBO reports. Thank you for your contribution to our journal.

Yours sincerely,
